# Aqueous Solution Equilibria and Spectral Features of Copper Complexes with Tripeptides Containing Glycine or Sarcosine and Leucine or Phenylalanine

Giselle M. Vicatos [1], Ahmed N. Hammouda [1,2], Radwan Alnajjar [1,2], Raffaele P. Bonomo [3], Gabriele Valora [3], Susan A. Bourne [1] and Graham E. Jackson [1,*]

[1] Department of Chemistry, University of Cape Town, Cape Town 7701, South Africa; VCTGIS001@myuct.ac.za (G.M.V.); ahmed.hammouda@uct.ac.za (A.N.H.); ALNRAD001@myuct.ac.za (R.A.); susan.bourne@uct.ac.za (S.A.B.)

[2] Department of Chemistry, Faculty of Science, University of Benghazi, Benghazi 16063, Libya

[3] Dipartimento di Scienze Chimiche, Università degli Studi di Catania, 95131 Catania, Italy; rbonomo@unict.it (R.P.B.); gabriele.valora@unict.it (G.V.)

\* Correspondence: graham.jackson@uct.ac.za; Tel.: +27-216-502-531

**Abstract:** Copper(II) complexes of glycyl-L-leucyl-L-histidine (GLH), sarcosyl-L-leucyl-L-histidine (Sar-LH), glycyl-L-phenylalanyl-L-histidine (GFH) and sarcosyl-L-phenylalanyl-L-histidine (Sar-FH) have potential anti-inflammatory activity, which can help to alleviate the symptoms associated with rheumatoid arthritis (RA). From pH 2–11, the MLH, ML, MLH$_{-1}$ and MLH$_{-2}$ species formed. The combination of species for each ligand was different, except at the physiological pH, where CuLH$_{-2}$ predominated for all ligands. The prevalence of this species was supported by EPR, ultraviolet-visible spectrophotometry, and mass spectrometry, which suggested a square planar CuN$_4$ coordination. All ligands have the same basicity for the amine and imidazole-N, but the methyl group of sarcosine decreased the stability of MLH and MLH$_{-2}$ by 0.1–0.34 and 0.46–0.48 log units, respectively. Phenylalanine increased the stability of MLH and MLH$_{-2}$ by 0.05–0.29 and 1.19–1.21 log units, respectively. For all ligands, $^1$H NMR identified two coordination modes for MLH, where copper(II) coordinates via the amine-N and neighboring carbonyl-O, as well as via the imidazole-N and carboxyl-O. EPR spectroscopy identified the MLH, ML and MLH$_{-2}$ species for Cu-Sar-LH and suggested a CuN$_2$O$_2$ chromophore for ML. DFT calculations with water as a solvent confirmed the proposed coordination modes of each species at the B3LYP level combined with 6-31++G\*\*.

**Keywords:** copper; speciation; equilibria; thermodynamic stability; peptides

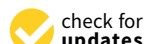



## 1. Introduction

Rheumatoid arthritis (RA) is a chronic, inflammatory, systemic and debilitating disease characterized by the destruction of diarthrodial joints. It is considered to be an autoimmune disease, and the exact cause is still unknown [1–3]. Often the symptoms are treated with steroidal and non-steroidal anti-inflammatory drugs, and sometimes surgical intervention is necessary in cases of severe joint deformation [4,5]. It is well known that copper(II) complexes have anti-inflammatory activity and cause a reduction in inflammation in a dose-dependent manner when administered subcutaneously [6–10]. Copper(II) exists in the plasma by binding non-reversibly to ceruloplasmin, reversibly to serum albumin, as well as by being distributed among low molecular mass ligands. These low molecular mass ligands have anti-inflammatory activity and are thought to be involved in transportation between cells in the body [11–13]. Therefore, in principle, an increase in the fraction of copper(II) complexes with low molecular mass ligands would lead to augmenting their anti-inflammatory activity when administered either orally or by subcutaneous injection [14].

Sorenson [15] and Jackson et al. [5,16] have investigated copper(II) complexes with low molecular mass ligands and have shown that they are effective in reducing the inflammation

associated with RA and have reduced toxicity. This suggests that the anti-inflammatory effect of endogenous copper(II) can be enhanced by exogenous sources. The preferred route of administration is dermal absorption as it is both harmless and convenient for patients.

In normal plasma, serum albumin is a major reversible, metal binding protein, accounting for 40 μg of copper(II) per ml of plasma [17,18]. The copper(II) binds to the amine, the amide and the imidazole of Asp-Ala-His in the C-terminus of the protein [19,20]. Using this sequence as a design template, Zvimba [5,18] and Odisitse [1,6,21,22] synthesized and tested a series of amide ligands. Zvimba [18] found that replacing an amine with an amide significantly reduced the in vivo copper(II) mobilizing ability of the ligand, but increased the lipophilicity of the complex. On the other hand, in murine biodistribution studies, Odisitse [22] found significant trans-dermal absorption and retention of copper(II) using amide ligands with a terminal pyridine.

Since peptides are a readily available source of diverse amides, in recent years research has centered around developing tripeptide ligands that would form a complex with copper(II) and studying their anti-inflammatory activity. For example, Grunchlik et al. [23] studied the role that the two tripeptides, Gly-His-Lys and Gly-Gly-His, had on skin inflammation, and concluded that copper(II) peptides could be used on skin as an alternative to corticosteroids or nonsteroidal anti-inflammatory drugs. Hostynek et al. [24] studied the tripeptide glycyl-L-histidyl-L-lysine cuprate diacetate and found that, through transdermal absorption, a potentially effective therapeutic amount of copper(II) complexed to the tripeptide was delivered to treat the inflammatory disease. Elmagbari [25] studied a series of tripeptides and found that the copper(II) complexes were hydrophilic. Specifically, the results from Vicatos [26] and Hammouda [27] were used to base the design of the tripeptides for this study.

Vicatos [26] studied two tripeptides, sarcosyl-L-leucyl-phenylalanine (Sar-Leu-Phe) and glycyl-L-leucyl-phenylalanine (Gly-Leu-Phe), and found that they were poor at mobilizing copper(II) in vivo, as they preferred to bind to zinc(II). Hammouda [27] studied a series of tripeptides, finding that the copper(II) mobilizing capacities were higher when histidine was in the third position, as opposed to having histidine in the second position, or if histidine is not present in the tripeptide. However, partition coefficient and membrane permeability studies showed that the copper(II) complexes of Vicatos' peptides were more lipophilic and were 2.1–8.8 times more permeable than the ligand complexes that Hammouda reported [27]. From the two studies of Hammouda [27] and Vicatos [26], it was concluded that an imidazole group in the third position should be included in the ligand design to ensure that the ligands would be selective for copper(II). The permeability coefficient measurements of Vicatos [26] suggest that the amino acid in position 2 of the tripeptide should be non-polar leucine or phenylalanine, and the first position of the ligand can either be glycine or sarcosine. N-methylated ligands have been shown to increase the lipophilicity and biological half-life of complexes compared to non-N-methylated ligands [28,29]. Thus, sarcosine was included in this study. The resultant ligands are therefore GLH, Sar-LH, GFH and Sar-FH.

## 2. Results and Discussions

### 2.1. Potentiometry

In an aqueous solution, GLH, Sar-LH, GFH and Sar-FH are zwitterions, which have three available sites for protonation. Glycine and sarcosine have a primary and secondary amine, respectively, while histidine has both a carboxyl group and an imidazole nitrogen. With decreasing pH, the amine group ($pK_a \approx 9.8$) will become protonated first, then the imidazole ring ($pK_a \approx 6.0$), and, finally, the carboxyl group ($pK_a \approx 1.8$). Their protonation constants ($\log \beta_{pqr}$) are reported in Table 1.

**Table 1.** Protonation and stability constants of GLH, Sar-LH, GFH and Sar-FH with copper(II). $\beta_{pqr} = [M_pL_qH_r]/[M]^p[L]^q[H]^r$, $I =$ in 0.15 mol.dm$^{-3}$ (NaCl), T = 25 °C.

| Ligand | p q r | log $\beta_{pqr}$ | Complex | p q r | log $\beta_{pqr}$ |
|--------|-------|-------------------|---------|-------|-------------------|
| GLH | 0 1 1 | 8.21 | Cu-GLH | 1 1 1 | 12.71 |
| | 0 1 2 | 15.10 | | 1 1 −1 | 2.77 |
| | 0 1 3 | 17.88 | | 1 1 −2 | −2.24 |
| Sar-LH | 0 1 1 | 8.45 | Cu-Sar-LH | 1 1 1 | 12.37 |
| | 0 1 2 | 15.32 | | 1 1 0 | 7.38 |
| | 0 1 3 | 18.05 | | 1 1 −2 | −2.70 |
| GFH | 0 1 1 | 7.95 | Cu-GFH | 1 1 1 | 12.76 |
| | 0 1 2 | 14.82 | | 1 1 −2 | −1.03 |
| | 0 1 3 | 17.65 | | | |
| Sar-FH | 0 1 1 | 8.22 | Cu-Sar-FH | 1 1 1 | 12.66 |
| | 0 1 2 | 15.09 | | 1 1 −2 | −1.51 |
| | 0 1 3 | 17.96 | | | |

　　　With the introduction of copper(II), these four ligands formed complexes, but each ligand formed a different set of species over a pH range from 2–11. The stability constants for the different species are seen in Table 1, and their distribution diagrams are seen in Figure 1. For these copper(II) complexes to satisfy the aim of undergoing transdermal absorption and releasing copper(II) ions into the blood plasma, it is essential to analyze their stability. It is noted that the complexes must be stable enough to form, but not so stable that copper(II) cannot be released once it is in the blood plasma.

　　　In particular, the stability between the N-methylated group on the complexes with sarcosine versus the non-N-methylated group on the complexes with glycine was analyzed. To do this, constants from GLH were compared with constants from Sar-LH and constants from GFH were compared with constants from Sar-FH. For the protonation constants, the species from Sar-LH or Sar-FH, i.e., with the methyl group, are 0.17–0.31 log units bigger and more stable than their glycine counterparts. For the copper(II) complexation stability constants, the MLH and MLH$_{-2}$ species without the methyl group (GLH and GFH) increased the stability by 0.1–0.34 log units and 0.46–0.48 log units, respectively, and therefore are more stable than the methylated species (Sar-LH and Sar-FH). The stability constants of the ML and MLH$_{-1}$ species could not be compared.

　　　The methyl group has an electron-donating inductive effect, and so it was expected that the ligands/complexes with sarcosine would have larger stability coefficients and thus be more stable than the complexes with glycine. This was seen for the protonation constants, and therefore it is confirmed that the methyl group does affect the stability. However, it was mostly found that the complexes without the methyl group were slightly more stable, which leads to the suggestion that the methyl group also has steric effects. A steric effect makes the inductive effect less prominent. Another reason could be due to the ammonium ions or charged amine groups preferentially forming hydrogen bonds with water and thereby decreasing the available charge. This phenomenon was also reported for solvated alkylamines, where the order of base strength was rearranged. The true base strength in a vacuum should have been $NH_3 < RNH_2 < R_2NH < R_3N$, since the methyl groups increase, which subsequently increases the electron density on the nitrogen atom. However, the order for aqueous solutions was found to be $NH_3 < RNH_2$, $R_2NH > R_3N$, which was explained by the preference of the amine to form hydrogen bonds with water [30].

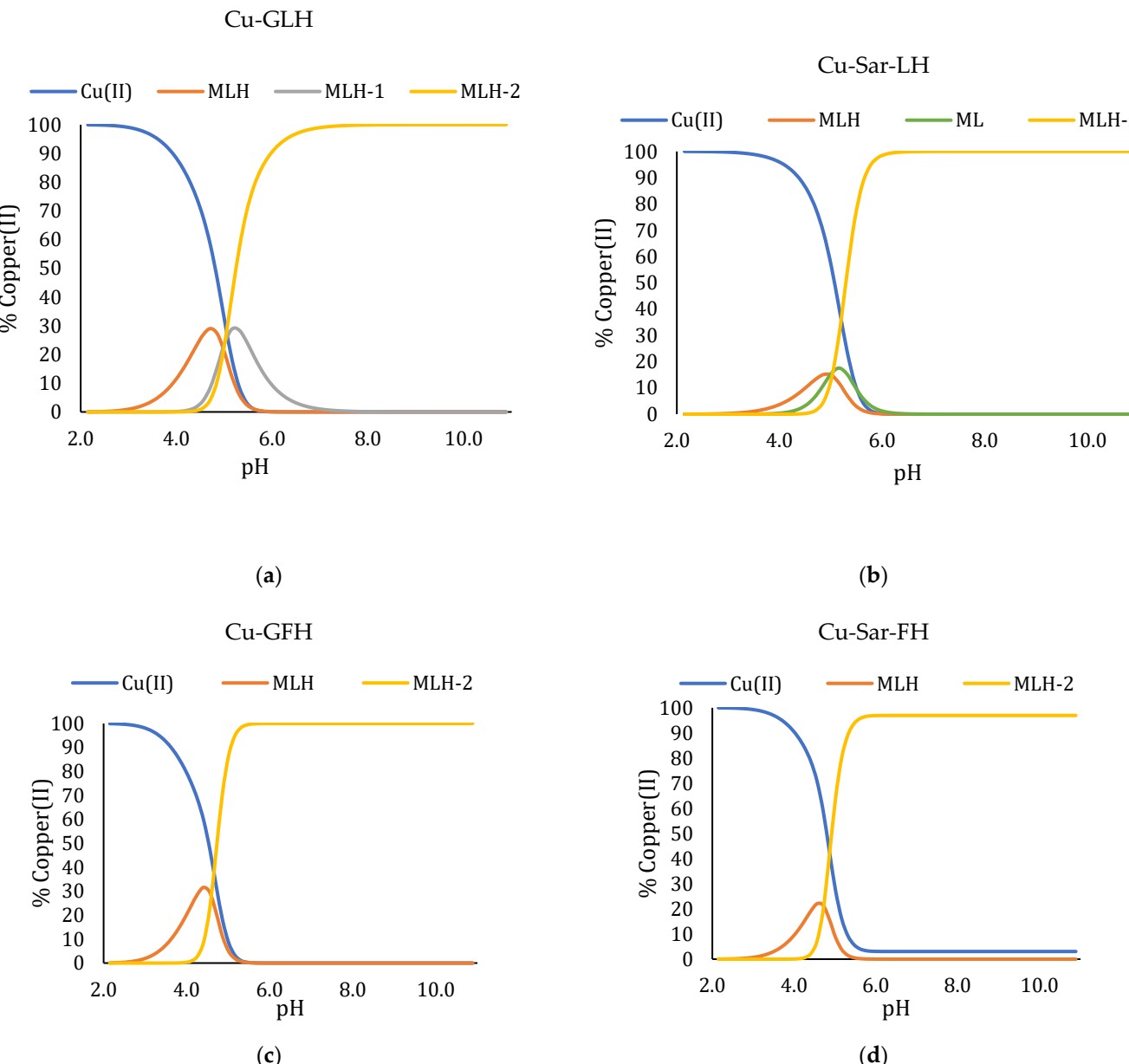

**Figure 1.** Protonation species distribution curve for copper(II) and (**a**) GLH (1:1 ratio), (**b**) Sar-LH (1:1 ratio), (**c**) GFH (1:1 ratio) and (**d**) Sar-FH (1:1 ratio) at 25 °C in 0.15 mol dm$^{-3}$ (NaCl).

An analysis of the leucine and phenylalanine amino acids can also be conducted to determine how they affect the stability constants. Thus, GLH and GFH are compared, and Sar-LH and Sar-FH are compared. Starting with the protonation constants, the species with leucine (GLH and Sar-LH) are 0.09–0.28 log units bigger and more stable than the species with phenylalanine (GFH and Sar-FH). For the copper(II) complexes, the MLH and MLH$_{-2}$ species with phenylalanine (GFH and Sar-FH) increased the stability by 0.05–0.29 log units and 1.19–1.21 log units, respectively. The stability constants for the ML and MLH$_{-1}$ species could not be compared.

The isobutyl group of leucine is electron donating, which should increase the basicity of the ligand. On the other hand, the benzyl group of phenylalanine is electron withdrawing, which should decrease the basicity of the ligand. This was seen in the protonation constants. However, when analyzing the metal stability constants, a similar scenario is seen as with the comparison between the metal complexes of glycine and sarcosine, where steric or entropy effects influence the stability of the metal complexes.

An indication of how the MLH species coordinates was found by comparing the stability constants of this species with the stability constants of other ligands found in the literature. The MLH species most likely has the copper(II) either coordinated to the amine and neighboring carbonyl-O, or to the imidazole-N and the carboxyl-O. For the former comparison, the literature compounds glycylglycine (GGOMe) and glycylsarcosine (GSOMe) were selected. They coordinate to copper(II) through the amine-N and carbonyl-O. Their log $K$ values are 4.11 and 5.18, respectively [31]. To compare these values with the MLH species of Cu-GLH, Cu-Sar-LH, Cu-GFH and Cu-Sar-FH, the corresponding protonation of the imidazole group had to be subtracted from the complex stability constant. This gives log $K$ values of 5.82, 5.5, 5.89 and 5.71, respectively. These values correspond closely to the literature values above, suggesting that this is the mode of coordination of these ligands. Unfortunately, the comparison could not be made for the imidazole-N and carboxyl-O, since literature for this comparison could not be found. However, after an ¹H NMR analysis (Section 3.4), there is reason to believe that copper(II) also coordinates to the imidazole-N and to the carboxyl-O. This "double-sided" coordination was also proposed for the MLH species of copper(II) complexes containing peptides with a histidyl residue [32]. Figure 2 shows the two coordination modes of MLH for GLH, which can also be transferred to Sar-LH, GFH and Sar-FH.

(**a**)                 (**b**)

**Figure 2.** The two coordination modes (**a**) with an amine-N and carbonyl-O coordination and (**b**) with an imidazole-N and carboxyl-O coordination for the MLH species of GLH.

The ML species of Cu-Sar-LH occurs at a higher pH than MLH, and so it is most likely that copper(II) only coordinates via the amine route and not via the imidazole ring, since the charge of the carboxylic acid moiety does not need to be neutralized at high pH values. Therefore, a possible coordination for the ML species is to the amine and neighboring amide-N with a protonated imidazole-N. The most likely mechanism in going from an MLH to ML species is for the sarcosine amide to switch from a carbonyl-O to an amide-N coordination, while the imidazole-N remains protonated. The transition from MLH to ML gives a p$K_a$ value of 4.99, which is a value typical of a metal-assisted amide deprotonation. The proposed switch has been seen in the ML to MLH$_{-1}$ species of Cu-di, tri and tetraglycine [33,34]. These ligands have p$K_a$ values of 4.23, 5.41 and 5.56, respectively, and represent the first amide deprotonation. Additionally, a Cu-GGG complex [34] showed that the neighboring carbonyl-O could also be involved in the coordination. Therefore, the two possible coordination modes for the ML species (Figure 3) could be a bidentate coordination consisting of the amine and neighboring amide-N with a protonated imidazole-N, or a tridentate coordination consisting of the amine, neighboring amide-N and the carbonyl-O, with a protonated imidazole-N.

**(a)**    **(b)**

**Figure 3.** The two coordination modes (**a**) with an amine-N and neighboring amide-N coordination and (**b**) with an amine-N, neighboring amide-N and carbonyl-O coordination for the ML species of Sar-LH.

The coordination mode of the $MLH_{-2}$ species cannot be determined using $pK_a$ values, but EPR and UV-vis have proposed the coordination to the amine-N, both amide-Ns and to the imidazole-N. This 4N-complex has been seen in other copper(II) peptide complexes [33,35]. The $pK_a$ value of the ML to $MLH_{-2}$ species for Cu-Sar-LH is not attainable, but it can be speculated that this transition happens in one step where another coordination switch occurs between the carbonyl-O of leucine and its neighboring amide-N, as well as the coordination to the imidazole-N. Likewise, the $pK_a$ values of the MLH to $MLH_{-2}$ species for Cu-GFH and Cu-Sar-FH are not attainable, but the coordination to two amide-Ns and to the imidazole-N probably occur in one step.

Similar to the ML species of Cu-Sar-LH, for the $MLH_{-1}$ species of Cu-GLH, copper(II) is also thought to coordinate only via the amine route. The $pK_a$ value for the $MLH_{-1}$ to $MLH_{-2}$ transition is 5.01 and was originally thought to suggest a second amide deprotonation. This would produce an $MLH_{-1}$ coordination mode where copper(II) is coordinated to the amine, neighboring amide-N and imidazole-N. Transitioning to the $MLH_{-2}$ species would then just include the coordination to the second amide-N. However, since the value of 5.01 represents the second amide deprotonation, when comparing this value with Cu-GGG ($pK_a$ = 6.86) [33,34], it is not close enough to confirm a second amide deprotonation. Two other $MLH_{-1}$ coordination possibilities are firstly to the amine and two amide-Ns with a protonated imidazole-N, or secondly to the amine, two amide-Ns and the carboxyl-O with a protonated imidazole-N. Transitioning to the $MLH_{-2}$ species would then include a coordination to the imidazole-N. However, the protonation of the imidazole-N has a log $K$ of 6.89, which is also not close enough to the 5.01 $pK_a$ value to confirm a coordination mode. All three coordination modes are reasonable and proposed as possible structures (Figure 4).

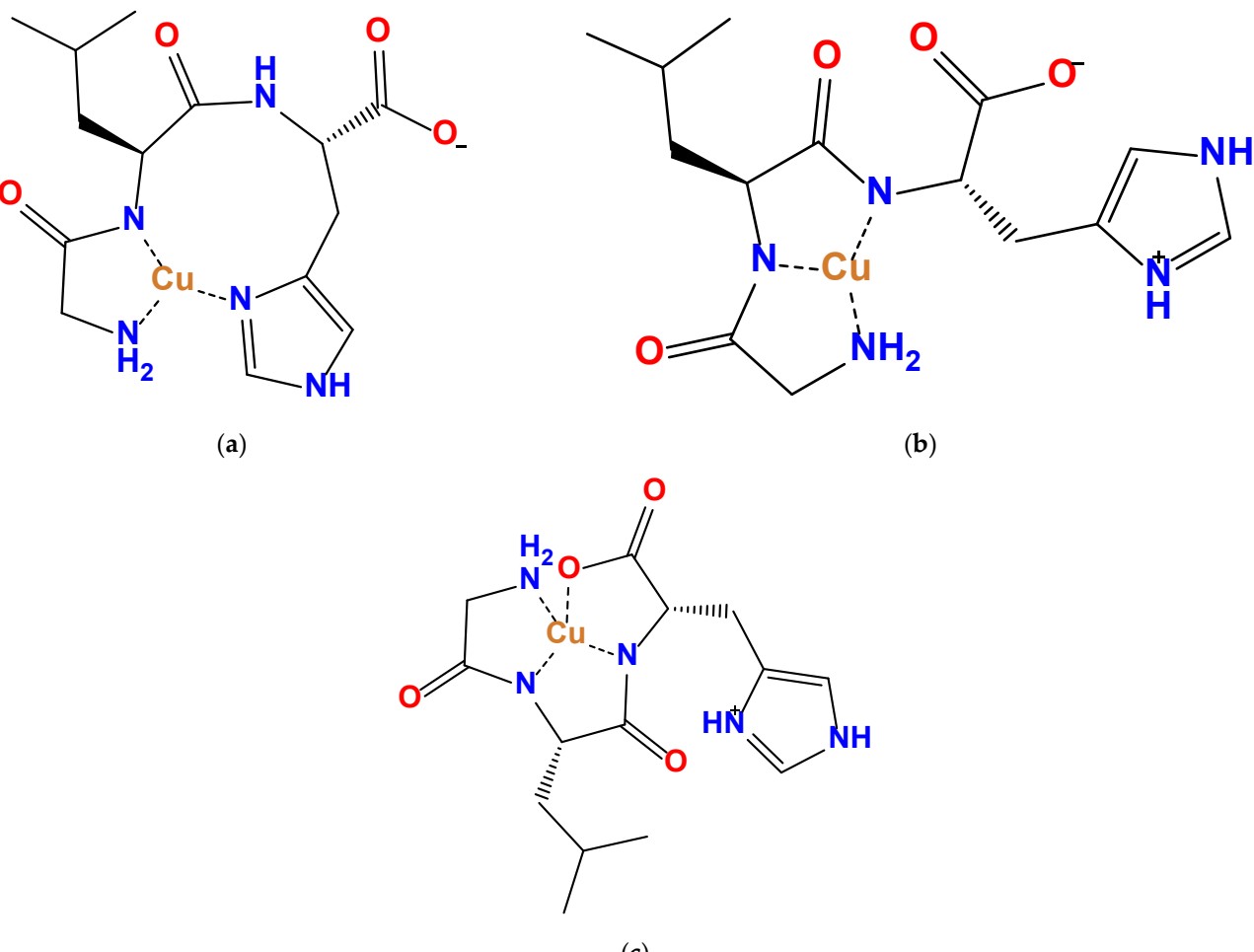

**Figure 4.** The three coordination modes (**a**) with an amine-N, neighboring amide-N and imidazole-N coordination, (**b**) with an amine-N and two amide-Ns coordination and (**c**) with an amine-N, two amide-Ns and carboxyl-O coordination for the $MLH_{-1}$ species of GLH.

### 2.2. Ultraviolet-Visible Spectrophotometry (UV-Vis)

Color changes occurred during the potentiometric titrations for the copper(II) complexes. As the pH of the copper(II) solution increased, its color changed from a clear to a violet-pink color. This color change showed the formation of species over the pH range 2–11 and aided in the identification of the species coordination modes [6]. The spectra of the four ligands (Figure 5) appear to be similar, since one absorption band with similar $\lambda_{max}$ values (Table 2) can be seen in all spectra.

**Table 2.** The maximum wavelengths and their corresponding molar extinction coefficients of the $MLH_{-2}$ species from Cu-GLH, Cu-Sar-LH, Cu-GFH and Cu-Sar-FH.

| Complex | $\lambda_{max}$ (nm) | $\varepsilon$ (dm$^3$ mol$^{-1}$ cm$^{-1}$) |
|---|---|---|
| Cu-GLH | 518 | 88 |
| Cu-Sar-LH | 523 | 105 |
| Cu-GFH | 517 | 85 |
| Cu-Sar-FH | 521 | 98 |

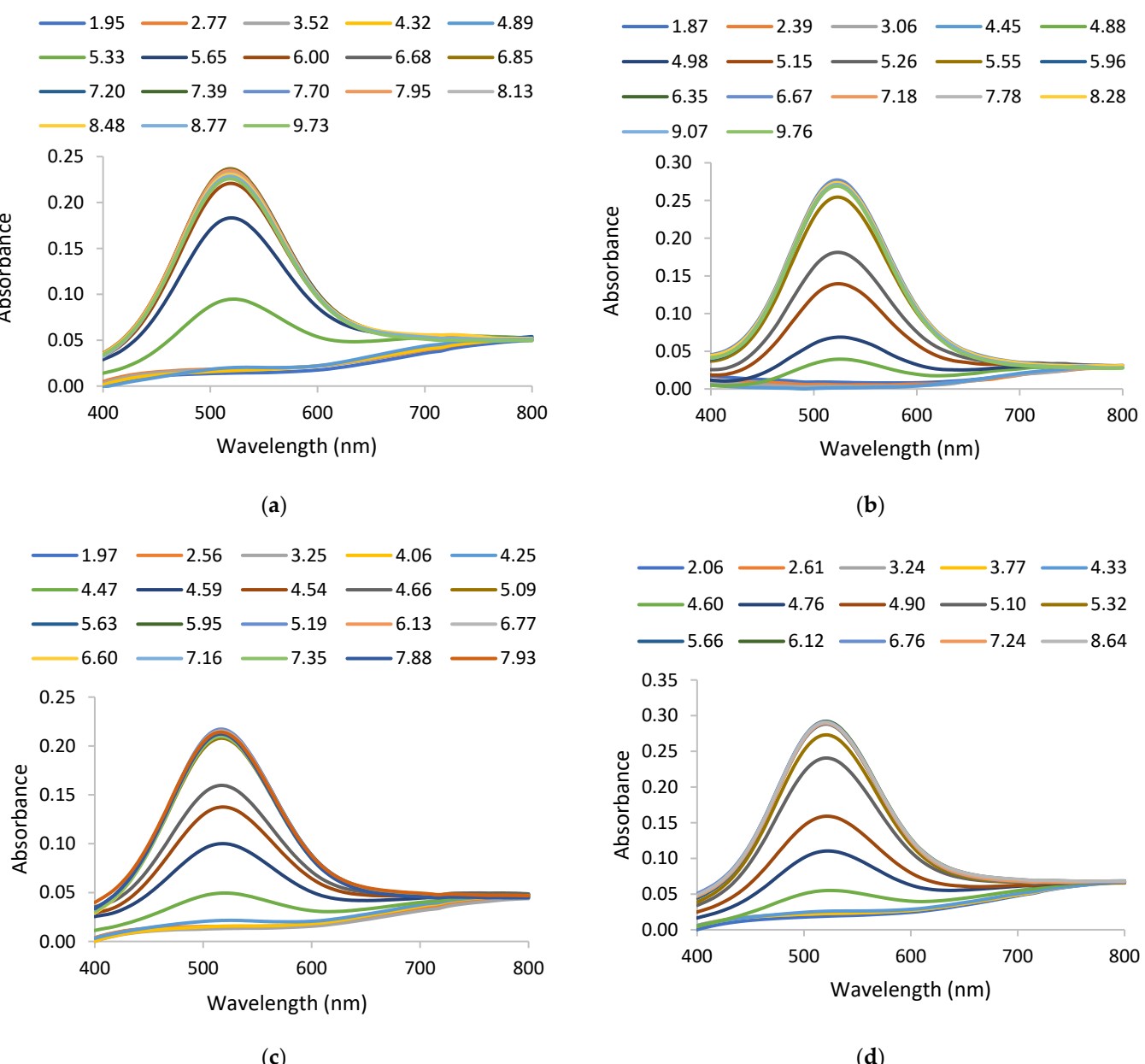

**Figure 5.** Electronic spectra of solutions containing (**a**) Cu-GLH ($3.04 \times 10^{-3}$ M of GLH and $2.69 \times 10^{-3}$ M of copper(II)), (**b**) Cu-Sar-LH ($2.93 \times 10^{-3}$ M of Sar-LH and $2.58 \times 10^{-3}$ M of copper(II)), (**c**) Cu-GFH ($3.09 \times 10^{-3}$ M of GFH and $2.55 \times 10^{-3}$ M of copper(II)) and (**d**) Cu-Sar-FH ($3.66 \times 10^{-3}$ M of Sar-FH and $2.97 \times 10^{-3}$ M of copper(II)).

Copper(II) ions have a $d^9$ configuration, which gives rise to Jahn–Teller distortion. This causes a lack of symmetry in an octahedrally coordinated copper(II) complex, which allows electron *d*–*d* transitions to occur and form an absorption band [36]. The four copper(II) complexes are expected to form tetragonally distorted octahedral complexes, which is expected to produce only a single absorption band in the visible region. Three spin-allowed transitions would occur, which are $^2A_{1g} \leftarrow {}^2B_{1g}$, $^2B_{2g} \leftarrow {}^2B_{1g}$ and $^2E_g \leftarrow {}^2B_{1g}$ [5,37], but because these bands are broad and cannot be distinguished, they appear as a single absorption band between 555 and 769 nm [37].

The absorption spectra in Figure 5 shows only one complex species ($\lambda_{max} = 517–523$ nm). For all the ligands, this complex species is the MLH$_{-2}$ species, since the $\lambda_{max}$ peaks follow the MLH$_{-2}$ formation trend. None of the other supposed species can be identified. Molar extinction coefficients for MLH$_{-2}$ (Table 2) were calculated by taking the absorbance band

produced at the highest pH and dividing it by the copper(II) concentration. Octahedral environments typically have extinction coefficients of about 10 dm$^3$ mol$^{-1}$ cm$^{-1}$ [36,38]. The higher molar extinction coefficients for the MLH$_{-2}$ species verify that the coordination of the complex is unsymmetrical. When using Sigel and Martin's [39,40] empirical equation and parameters, the equatorial coordination to the amine-N, two amide-Ns and an imidazole-N gives a calculated $\lambda_{max}$ of 531 nm. This is close to the observed $\lambda_{max}$ range of 517–523 nm [41,42] and is thus reasonable to suggest a square planar geometry with a CuN$_4$ coordination.

### 2.3. Electron Paramagnetic Resonance (EPR) Measurements

The four copper (II) complexes, Cu-GLH, Cu-Sar-LH, Cu-GFH and Cu-Sar-FH, were examined at pH 7, and following the species distribution diagrams, only one copper(II) species should be found at this pH. At first, the RT spectra were run in a more acidic pH to check the presence of prominent complex species in the system, which took into account that all the EPR spectra were run in the absence of added ionic strength.

The isotropic EPR spectrum is obtained at RT conditions, which is characterized by four lines (copper nuclear spin equal to 3/2, and then 2I + 1 lines). In contrast, an anisotropic EPR spectrum is obtained at LT conditions, which generally has an axial symmetry, characterized by two sets of four transitions occurring at different values of the magnetic field. These two sets are either parallel or perpendicular transitions. The parallel transitions of the LT EPR spectrum are well separated by the hyperfine coupling constant, whose values, together with the g parallel values, give information on the geometry of the copper(II) complex. By contrast, the perpendicular hyperfine coupling constant has a much lower numeric value than the parallel one, which is not resolved and presented as a single transition at higher magnetic fields. The two sets of transitions can overlap, and because the parallel lines are easily recognizable due to their low intensity, two or three of the parallel lines are generally visible.

The RT spectra of the four copper(II) complexes at pH 7 are reported in Figure 6. Looking at these spectra, it is evident that they all look very similar to each other. This can be a result of the MLH$_{-2}$ species, which is the predominant species in the pH range 4.5–11. Due to the presence of donor nitrogen atoms on the ligand at a higher magnetic field, it is possible to see that all the RT spectra contain the fourth line. This shows a superhyperfine (shf) structure that is coming from the delocalization of the spin density of the copper(II) free electron on the nitrogen atoms. Since the nitrogen nuclear spin is 1, from the multiplicity of the 2ΣI + 1 lines, it is possible to count the number of nitrogen atoms bound to copper(II) in the complex equatorial plane. Nine shf lines indicate that copper(II) is bound to four quasi-equivalent nitrogen atoms, and so the chromophore of this species can be considered as CuN$_4$.

The magnetic parameters from the LT EPR spectrum can be seen in Table 3. All the LT frozen EPR spectra show g$_{||}$ > g$_\perp$ ≥ 2.04 [43]. This suggests that the copper(II) ground state could reasonably be assigned to the d$_{x^2-y^2}$ orbital in an octahedral, square planar or square-based pyramidal geometry. The low value of the g$_{||}$ and the relatively high absolute value of the parallel hyperfine constant suggest that these copper(II) complex species have a square planar geometry in a probable macrochelate complex. Since only subtle and negligible differences are observed among the copper(II) complexes with these ligands, they can be considered to have the same stereochemistry.

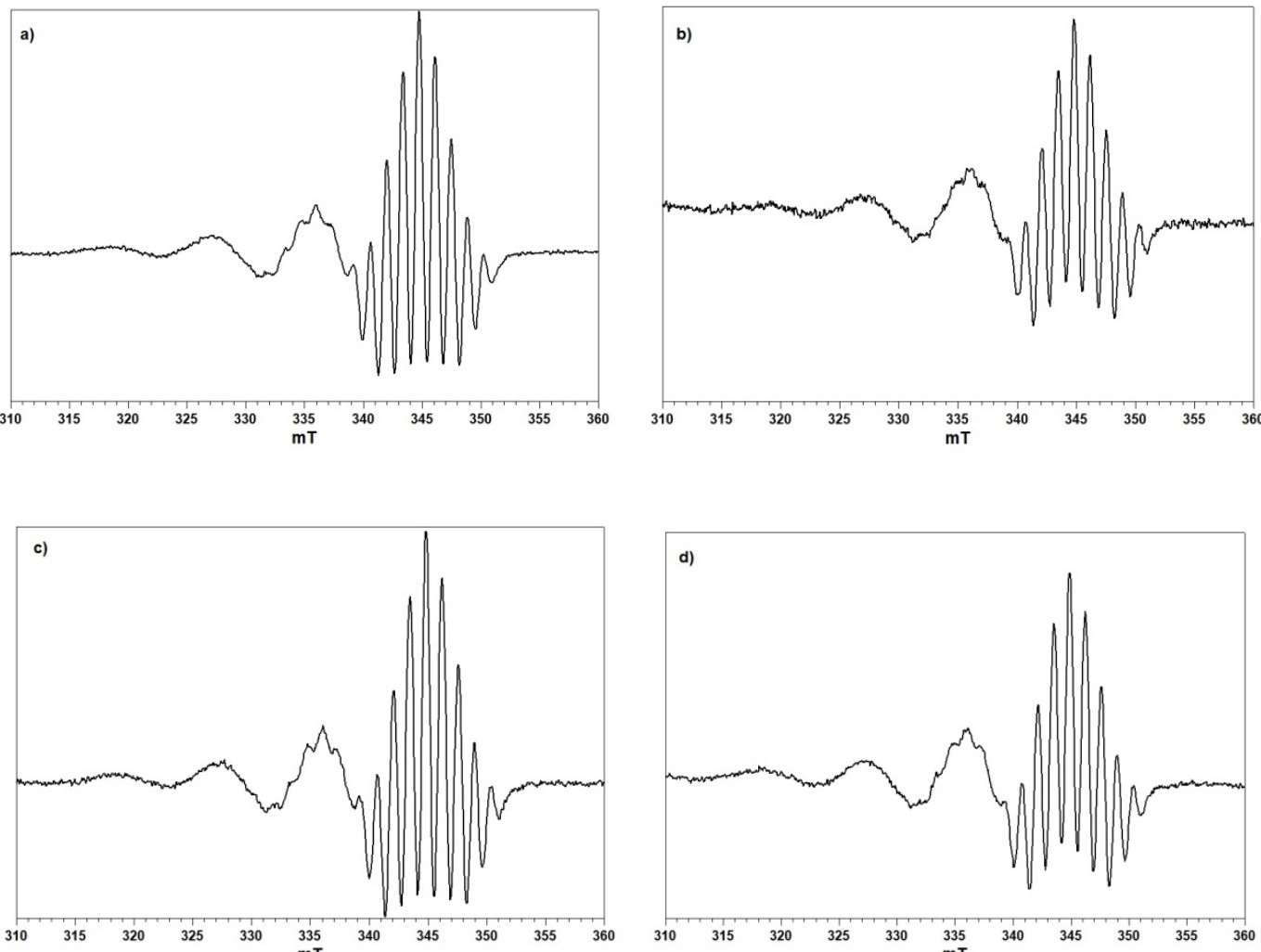

**Figure 6.** RT EPR 2nd derivative spectra recorded in aqueous solution for the copper(II) complexes, (**a**) Cu-GLH, (**b**) Cu-Sar-LH, (**c**) Cu-GFH and (**d**) Cu-Sar-FH.

**Table 3.** Spin Hamiltonian parameters of copper(II) complexes with Cu-GLH, Cu-Sar-LH, Cu-GFH and Cu-Sar-FH at pH 7.0, which have been drawn out from RT EPR spectra and LT frozen aqueous solution EPR spectra. All the hyperfine coupling constants are expressed in $10^4$ cm$^{-1}$ units. Presumed errors in the last decimal figure are reported between brackets.

| Complex | $g_{iso}$ (3) | $a_{iso}$ (3) | $g_{||}$ (4) | $A_{||}$ (4) | $g_{\perp}$ (7) | $A_{\perp}$ (7) | $a_{iso}^N$(1) | $A_{\perp}^N$ (1) | $A_{||}^N$ (1) |
|---|---|---|---|---|---|---|---|---|---|
| Cu-GLH | 2.092 | 86 | 2.174 | 210 | 2.046 | 23 | 14 | - | 15 |
| Cu-Sar-LH | 2.094 | 88 | 2.171 | 211 | 2.046 | 24 | 14 | - | 15 |
| Cu-GFH | 2.091 | 84 | 2.175 | 208 | 2.040 | 27 | 14 | 11 | 16 |
| Cu-Sar-FH | 2.091 | 86 | 2.172 | 208 | 2.040 | 28 | 14 | - | 15 |

When lowering the pH of the aqueous solutions to pH 5, only Cu-Sar-LH displays signals from other species that are present in the system. The RT spectrum showing three different complex species of Cu-Sar-LH at pH 5.1 can be seen in Figure 7.

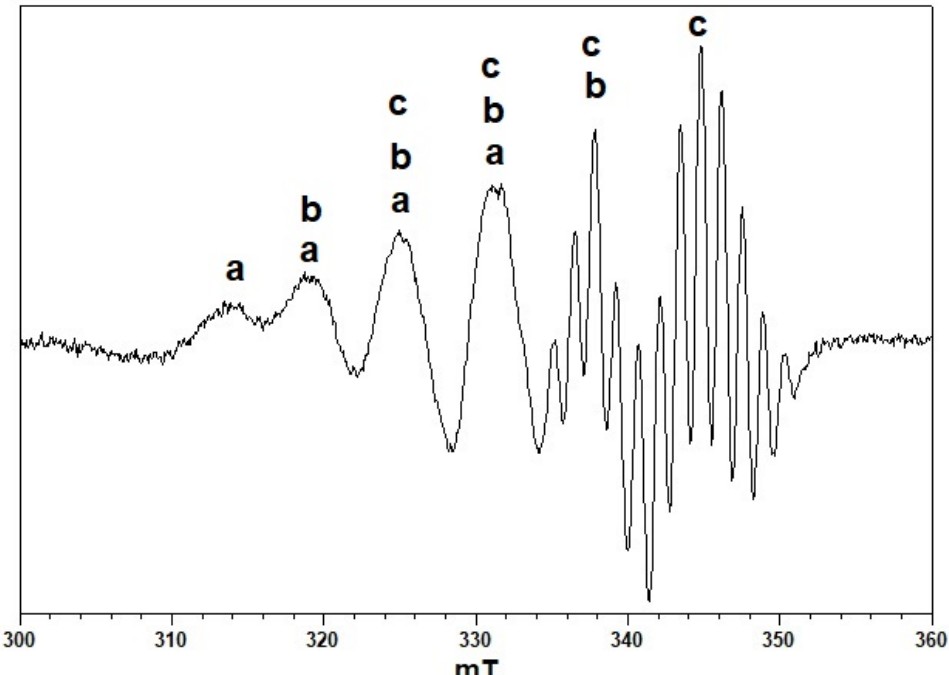

**Figure 7.** RT EPR 2nd derivative spectrum in an aqueous solution of Cu-Sar-LH at pH 5.1. Three copper(II) species are found and are designated as **a**, **b** and **c**.

Species **c** is the same species that was found at higher pH values and is thus the $MLH_{-2}$ species. It can also be noted that it is evident that the $MLH_{-2}$ species has already formed at pH 5.1. Species **a** and **b** both occur simultaneously with species **c** at this pH and give the following isotropic magnetic values: where **a** gives $g_{iso}$ = 2.151 ± 0.007 and $a_{iso}$ = 59 ± 0.005 × 10$^{-4}$ cm$^{-1}$ and **b** gives $g_{iso}$ = 2.111 ± 0.007, $a_{iso}$ = 62 ± 0.005 × 10$^{-4}$ cm$^{-1}$ and $a_{iso}^{N}$ = 13 ± 0.002 × 10$^{-4}$ cm$^{-1}$. Species **b** shows an shf interaction with a pattern of five lines and an approximate intensity distribution of 1:2:3:2:1, and thus this indicates a $CuN_2O_2$ chromophore. The last two lines of species **b** overlap with the shf lines of species **c**. Looking at the isotropic magnetic parameters, which show a low value of $g_{iso}$ and a high value of $a_{iso}$, it is probable that the pattern (species **b**) is due to the nitrogen atoms coming from the amine and neighboring amide-N. At pH 5.1, the imidazole-N is still protonated, and therefore the species would be the ML species. Species **a** shows isotropic magnetic parameters, which are compatible with the formation of MLH.

Unfortunately, the RT EPR spectra of Cu-GLH, Cu-GFH and Cu-Sar-FH were not well resolved in the pH range of 4.8–5.2. This meant that it was not possible to determine isotropic magnetic parameters for other species that could be present.

For all four complexes, the shf structure at LT is present in all the EPR spectra of the frozen solutions (pH 7–8). Unfortunately, it is only resolved in the parallel part of the spectrum for Cu-GFH (Figure 8). This LT spectrum has the nine-line shf structure that was already seen in the RT EPR spectra in Figure 6.

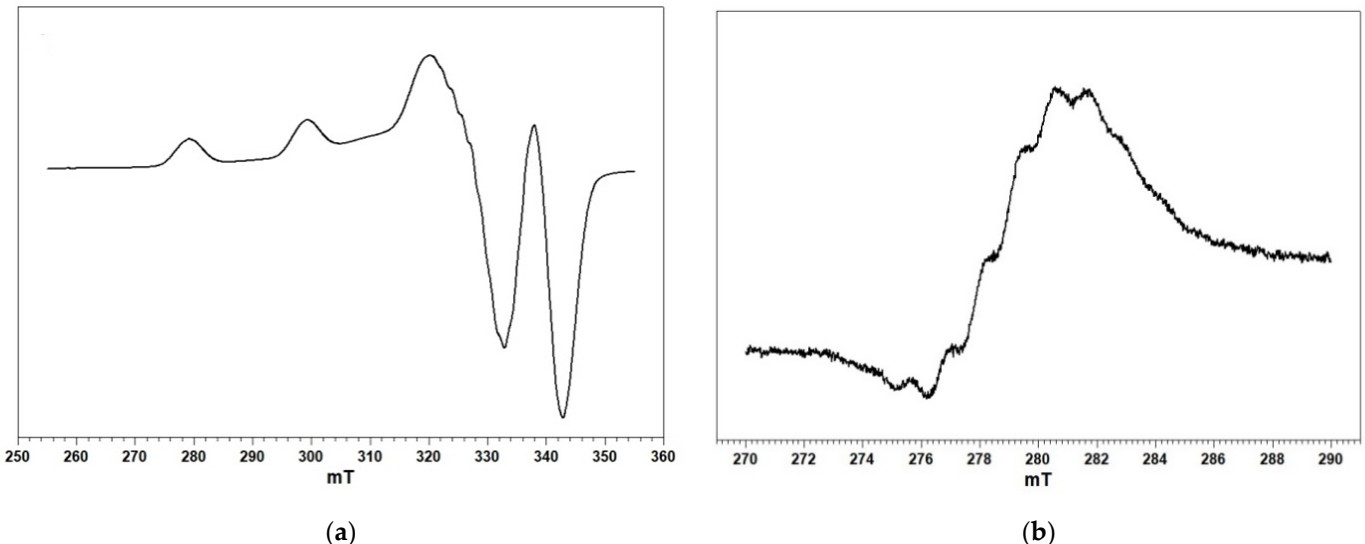

(**a**)                                                        (**b**)

**Figure 8.** (**a**) LT EPR spectra of Cu-GFH in frozen aqueous solution at pH 7–8 and (**b**) LT EPR 2nd derivative spectrum of the lowest magnetic field feature of Cu-GFH in frozen aqueous solution at pH 7–8.

### 2.4. H NMR Spectroscopy

Copper(II) is a paramagnetic metal ion that affects both the chemical shifts and relaxation rates of the ligand nuclei. This is due to the strong interactions between the unpaired electron of copper(II) and the magnetic dipoles of the nuclei of the ligand [44,45]. These interactions can be described as a through-bond effect (Fermi contact interaction) and a through-space effect (dipolar interaction). As a result, [1]H NMR can be used to determine the binding sites of copper(II) to the ligands. As copper(II) coordinates to the ligand, the dipolar interaction causes the relaxation rates to increase, and that produces broadening in the [1]H NMR signals. The increase in the relaxation rates are dependent on the distance between the nuclei of the ligand and the copper(II) ion, and so the closer the signals are to the binding sites, the more they broaden [46–51]. To be exact, the transverse relaxation time ($T_2$) determines the line width of the peaks, which is defined as:

$$T_2 \; = \; \frac{1}{\pi \, W_{1/2}} \qquad (1)$$

where $W_{1/2}$ is the line width at half height [52].

The technique to view the broadening of the [1]H NMR signals is to titrate the ligands with copper(II) at a predetermined pH. This will cause the [1]H NMR signals to differentially broaden, which is visually convenient for structural analysis. This is possible because NMR is a relatively slow spectroscopic technique, where the relaxation rates are in minutes, while the reactions for copper(II) exchange are fast. This means that an average is seen between the free and bound ligand spectra, which consequently causes the signals to broaden only gradually [48].

Figure 9 shows the [1]H NMR peak to proton assignments of the four ligands, as well as their change in chemical shifts for selected protons, as a function of pH. From the inflection points of these curves, the protonation stepwise formation constants can be estimated. The estimated p$K_a$ for L→LH is 8.8 for GLH, Sar-LH and Sar-FH, and 8.6 for GFH, and the estimated p$K_a$ for LH→LH$_2$ is 6.8 for GLH and Sar-LH, and 6.9 for GFH and Sar-FH. These estimated stepwise formation constants agree with the potentiometric results. The full [1]H NMR spectra for the four ligands are given in the Supplementary Information (Figure S1).

For all the ligands, the amide proton peaks **d** and **g/i**, as well as the proton signals with the assignments, **c** and **e**, were indistinguishable from each other, and so proton–proton correlation spectra (1D-TOCSY) (Figure S2) were collected to differentiate between the

signals. The results for GLH are given in Figure S2. The irradiating peak **d** (Figure S2a) affects peaks **b** and **b′**, which means peak **d** belongs to the amide-NH of the histidine amino acid. Peak **i** should then belong to the leucine amino acid, and as expected, by irradiating peak **i** (Figure S2b), peaks **b, b′** and **j** disappear, while peaks **h, f, f′** and **g** are affected. It is noted in both spectra that peaks **a** and **a′** are also affected; this is because they overlap the amide peaks and therefore have the same frequency as the amide groups. Figure S2c shows the expanded region of Figure S2b that contains peaks **c** and **e**; since peak **i** was irradiated and the second peak is affected, this means that this is peak **e**.

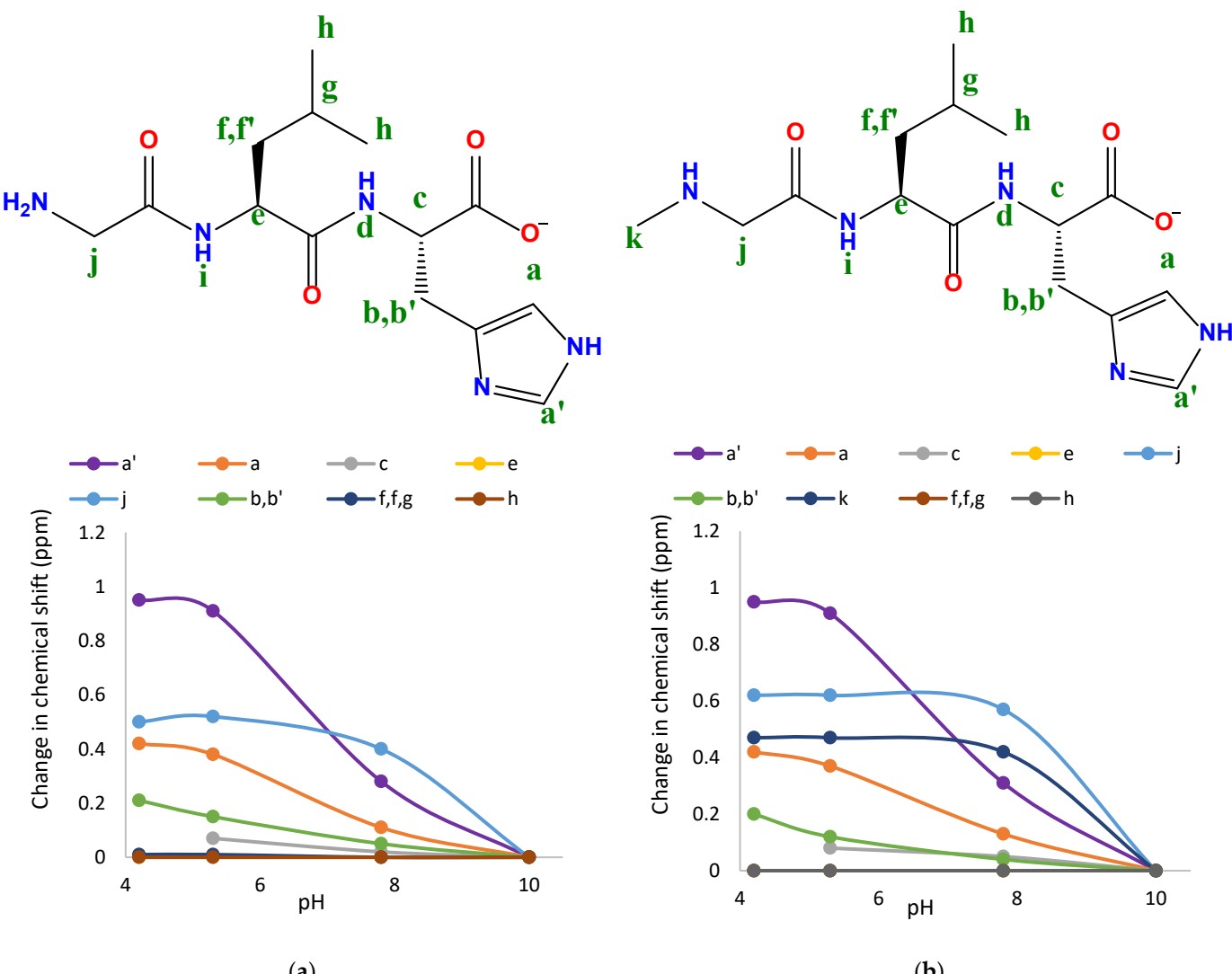

(**a**)

(**b**)

**Figure 9.** *Cont.*

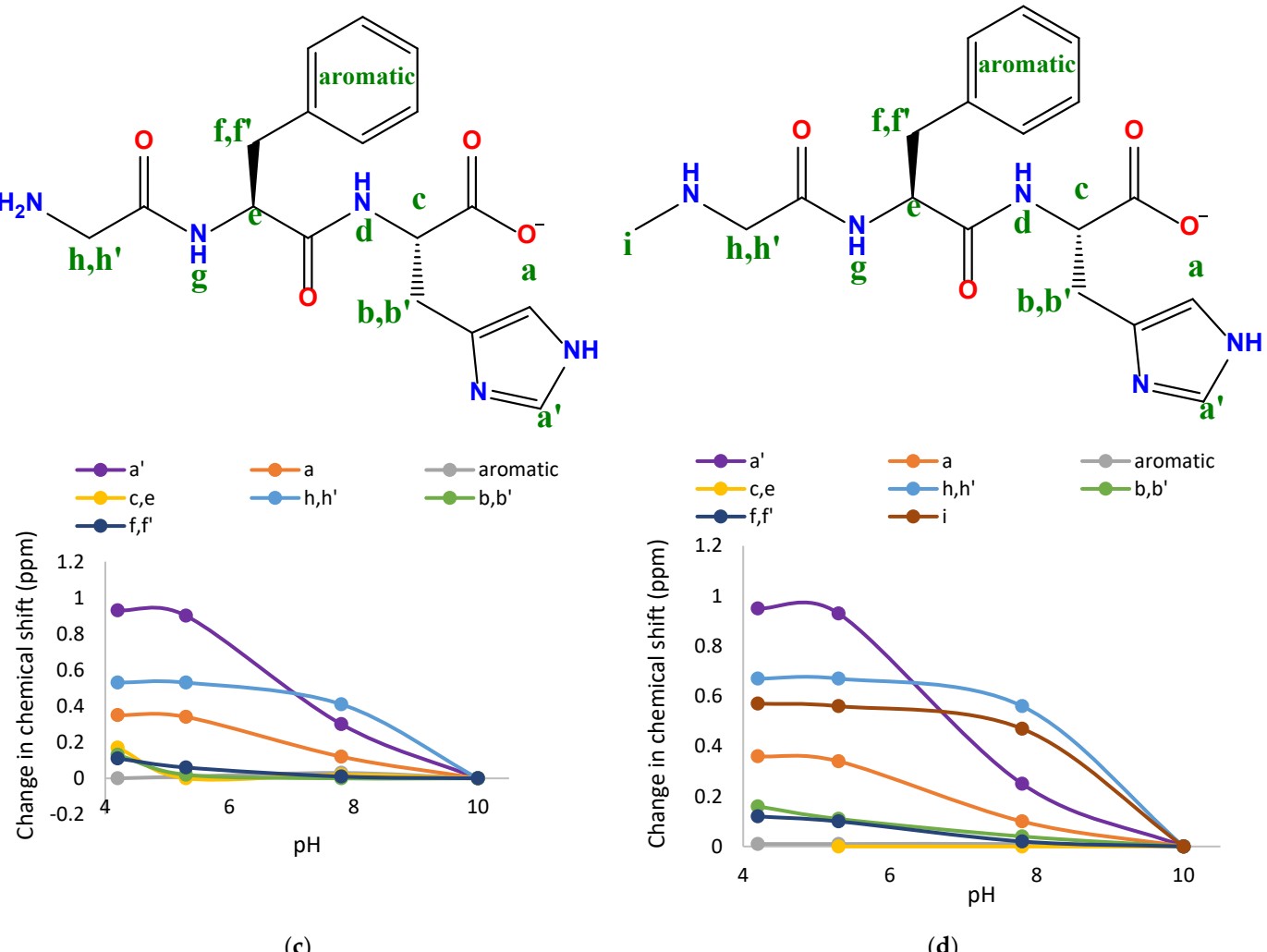

**Figure 9.** The $^1$H NMR proton assignments and change in $^1$H chemical shift as a function of pH for (**a**) GLH, (**b**) Sar-LH, (**c**) GFH and (**d**) Sar-FH.

All four copper(II) complexes form the MLH species, which, according to the species distribution diagrams, is the only species present at pH 3.5. Significant signal broadening will occur in the protons that are on carbons that are α or β to the coordination site, while other proton signals will not be affected to the same extent. The spectrum of Cu-GFH, as well as the spectrum of Cu-Sar-FH, have overlapping **b**, **b'**, **f** and **f'** peaks. By setting the pH to 4.8 and overlapping the spectra of GFH, Cu-GFH and GLH (Figure S3), as well as overlapping the spectra of Sar-FH, Cu-Sar-FH and Sar-LH (Figure S4), it was possible to distinguish between these peaks. At pH 4.8, the species present will cause peaks **b** and **b'** to broaden, while peaks **f** and **f'** will not broaden. Both GLH/Sar-LH and GFH/Sar-FH have peaks **b** and **b'** positioned at approximately 3 ppm, but only GFH/Sar-FH has peaks **f** and **f'** also positioned at approximately 3 ppm. As GFH/Sar-FH is titrated with copper(II), the comparisons show that only peaks **b** and **b'** broaden significantly.

Figure 10 shows the effect copper(II) has on the line width of the spectra of the four ligands at pH 3.5. For all ligands, the broadening of peaks **a**, **a'**, **b**, **b'** and **c** shows coordination to the imidazole-N and carboxyl-O. Peak **j** in the GLH spectra, peaks **j** and **k** in the Sar-LH spectra, peaks **e**, **h** and **h'** in the GFH spectra, and peaks **e**, **h**, **h'** and **i** in the Sar-FH spectra also broaden, which means that a second coordination to the amine-N and neighboring carbonyl-O also occurs. This "double-sided" coordination has also been suggested in the literature [32]. Note, at this pH, the amide proton signals of Sar-FH and GFH are still present, which means that the amide-N cannot be coordinated.

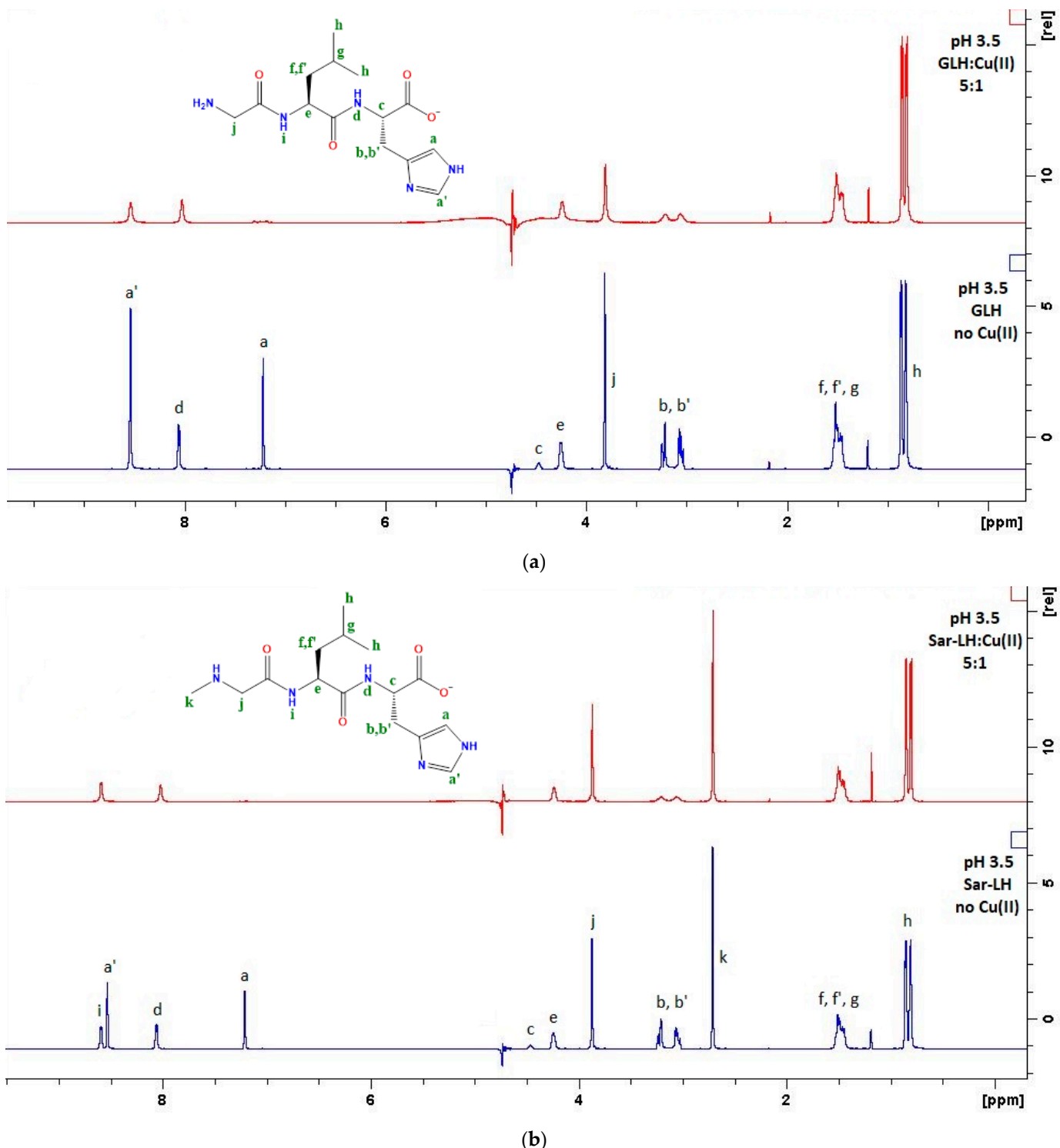

**Figure 10.** *Cont.*

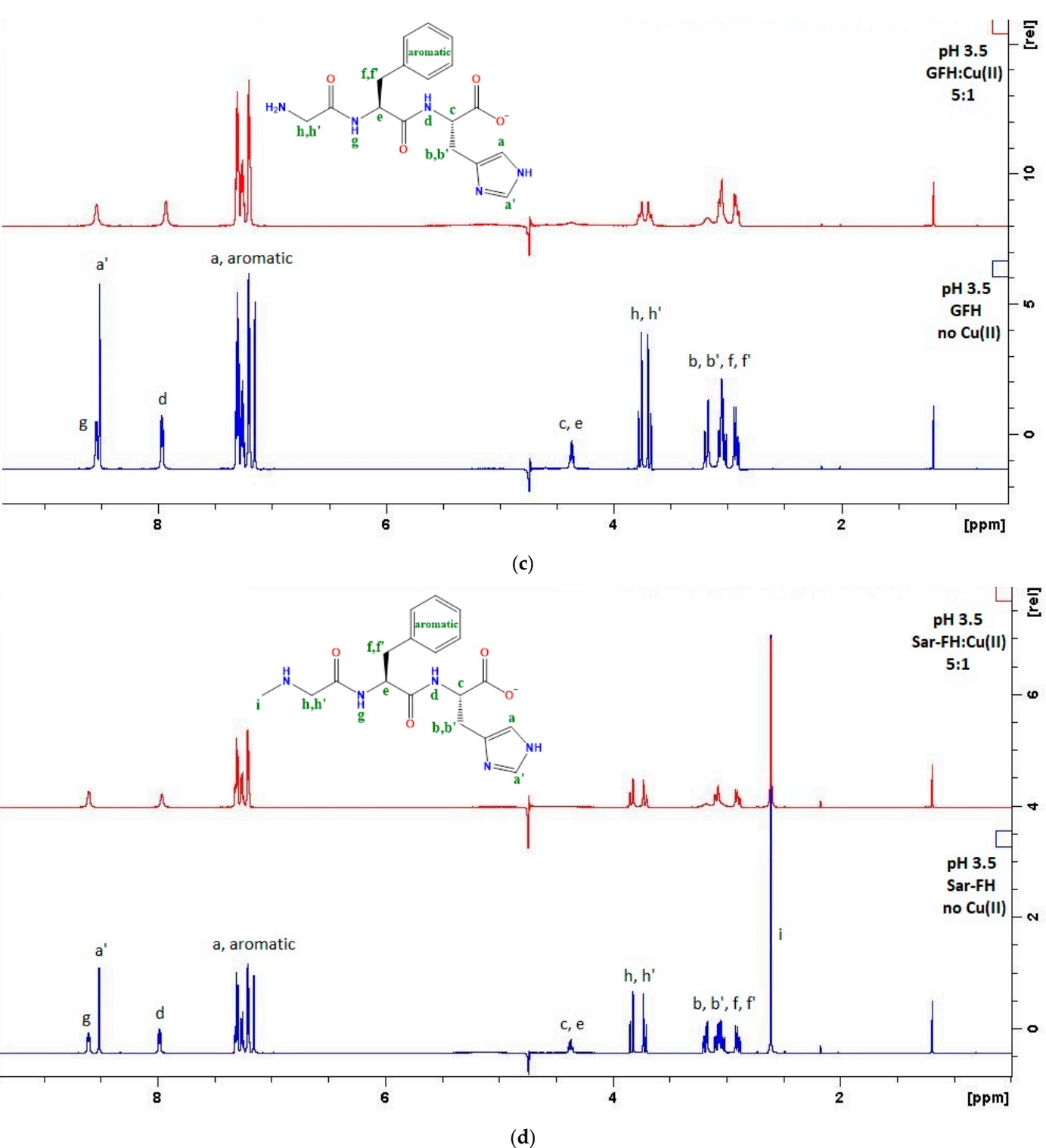

**Figure 10.** The $^{1}$H NMR spectra (blue) of the ligand (**a**) GLH, (**b**) Sar-LH, (**c**) GFH and (**d**) Sar-FH and the $^{1}$H NMR spectra (red) after (**a**) GLH, (**b**) Sar-LH, (**c**) GFH and (**d**) Sar-FH had been titrated with copper(II) to reach a 5:1 ligand:copper(II) ratio at a pH of 3.5 in 90% water and 10% D$_2$O.

Identifying the binding sites of the MLH$_{-1}$ species for Cu-GLH and the ML species for Cu-Sar-LH is not possible because the formation of these species overlaps with other species. At high pH values where the MLH$_{-2}$ species is the only present species, the peaks sharpen, which is due to the decrease in the exchange rate. Only the spectrum of the free ligand is seen, because the line widths of the complex are so broad (~457 Hz) [53] that

the spectrum will disappear into the baseline. This then results in the peaks appearing to sharpen instead of broaden, since only the spectrum of the free ligand is seen [54,55].

### 2.5. Mass Spectrometry

Since the copper species present in this study are all labile, the soft ionization, electrospray ionization mass spectrometry (ESI-MS) technique was used [39,56,57]. While this technique generally does not cause fragmentation, radical generation can lead to fragmentation [58]. Thus, a combination of intact and fragmented complexes is expected.

Since all four ligands behaved similarly, only the Sar-LH system will be discussed. In the positive mode spectrum (Figure 11a), the base peaks 340.10 $m/z$, 362.08 $m/z$ and 378.11 $m/z$ represent the uncomplexed ligand in the form of $(LH + H)^{1+}$, $(LH + Na)^{1+}$ and $(LH + K)^{1+}$, respectively. $K^+$ ions were not added during the preparation of the complex, and so they most likely come from residual salts in the injector or tubing of the mass spectrometer. The 4N coordination of the $MLH_{-2}$ species is verified by the peaks between 401.00–405.13 $m/z$. These peaks represent the overlap of the two different structures of the $MLH_{-2}$ species, namely, where copper is in the Cu(II) form and where it is reduced from Cu(II)→Cu(I). The reduction of the Cu(II) occurs by collision-induced processes in the medium vacuum area of the source, which causes "inner-sphere" ligand to metal electron transfer, as well as the de-coordination of odd-electron species [59,60]. For simplicity, when copper(II) is reduced to copper(I), the symbol "M" in the complex species will be labelled as "$M^I$", while copper(II) will remain as "M". The base peak at 401.00 $m/z$ represents the $MLH_{-2}$ species in the $(MLH_{-2} + 2H)^{1+}$ form, and the base peak at 402.05 $m/z$ represents the $(M^ILH_{-2} + 3H)^{1+}$ form. The base peak at 355.08 $m/z$ also could represent the $MLH_{-2}$ species, but after it has undergone fragmentation. A suggestion for fragmentation is $(MLH_{-2}$-carboxyl group$)^{1+}$. Structural assignments can be seen in Table 4. A similar scenario was seen for the other ligand complexes, and their structural assignments are given in Table S1. Note that, for Cu-GLH in Figure 11b, the base peak at 387.07 $m/z$ could represent either the $MLH_{-1}$ or $MLH_{-2}$ species, or that the peak is a combination of the two species in the form of $(MLH_{-1} + H)^{1+}$ and $(MLH_{-2} + 2H)^{1+}$, respectively. Similarly, the base peak at 388.05 $m/z$ could represent either the $MLH_{-1}$ or $MLH_{-2}$ species, or again a combination of both species in the form of $(M^ILH_{-1} + 2H)^{1+}$ and $(M^ILH_{-2} + 3H)^{1+}$, respectively. Additionally, note that, in Figure 11c, the base peak at 368.03 $m/z$ for Cu-Sar-FH represents the free ligand after it has undergone fragmentation. A suggestion for how the ligand has undergone fragmentation involves the decomposition of the phenyl moiety. This has been seen in the literature where tropone first loses CO to yield a phenyl cation radical, which then leads to the decomposition of the phenyl moiety [61].

**Table 4.** Structural assignments of $m/z$ base peaks that were found in the ESI-MS spectrum for Cu-Sar-LH at pH 5 (positive mode) with a 1:1 ratio and concentration of 1 mM for Sar-LH and 0.7 mM for copper(II) in aqueous solution.

| Complex | $m/z$ | Assignment |
|---|---|---|
| | 340.10 | $(LH + H)^{1+}$ |
| | 355.08 | $(MLH_{-2}$-carboxyl group$)^{1+}$ |
| | 362.08 | $(LH + Na)^{1+}$ |
| Cu-Sar-LH | 378.11 | $(LH + K)^{1+}$ |
| | 401.00 | $(MLH_{-2} + 2H)^{1+}$ |
| | 402.05 | $(M^ILH_{-2} + 3H)^{1+}$ |

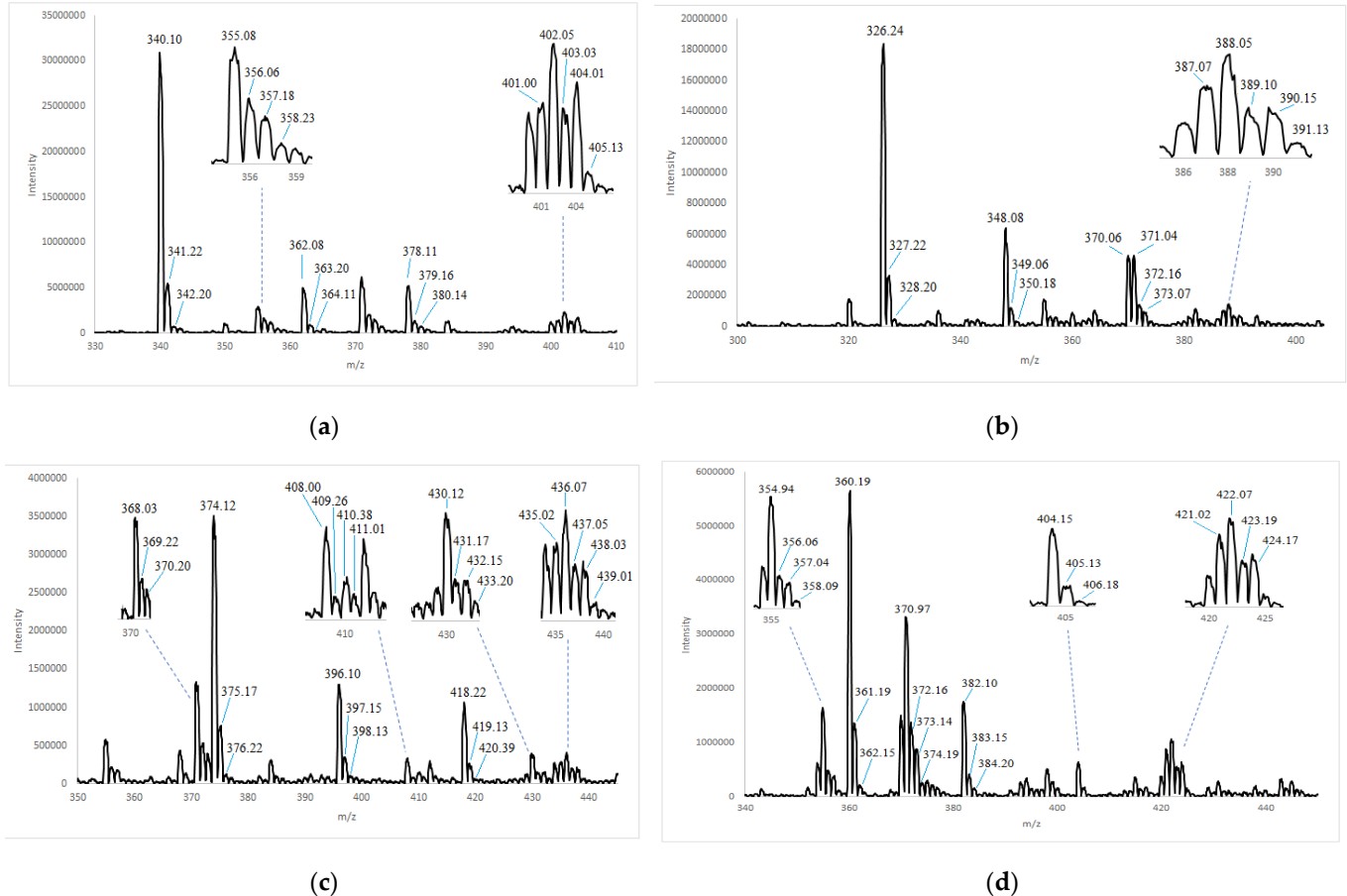

**Figure 11.** Section of the ESI-MS spectrum (positive mode) for the (**a**) Cu-Sar-LH, (**b**) Cu-GLH, (**c**) Cu-Sar-FH and (**d**) Cu-GFH complexes at a ratio of 1:1 and concentration of 1 mM for Sar-LH, GLH, Sar-FH and GFH, and 0.7 mM for copper(II) in aqueous solution at pH 5.

The $MLH_{-2}$ species is the dominant species for all the ligands, and therefore the identification of this species in the ESI-MS spectrum at pH 5 was relatively easy. The challenge was to try to identify the MLH species, as well as the ML species of Cu-Sar-LH and the $MLH_{-1}$ species of Cu-GLH, since they are present at much lower concentrations. The MLH and ML species could not be found. The assignment of peaks to the $MLH_{-1}$ species was debatable, and so the identification of this species was unconfirmed.

### 2.6. DFT Calculations

Molecular modelling using DFT calculations was used to validate the proposed structures and coordination modes for the $MLH_{-2}$ species. It was also used to indicate which coordination modes would most likely form in solution for the MLH, ML and $MLH_{-1}$ species, as well as to calculate $\lambda_{max}$. All starting structures had a copper(II) coordinate in an octahedral manner, with water molecules taking up vacant sites, and then they were optimized. If water molecules were moved out of the coordination distance to the copper atom after optimization, the resulting structures became a benchmark for additional optimization. The resultant structures for the MLH, ML, $MLH_{-1}$ and $MLH_{-2}$ species can be seen along with their geometry, the DFT calculated $\lambda_{max}$ value, and in the case of $MLH_{-2}$, the experimental $\lambda_{max}$ value, in Figures 12–15 respectively. All the possible coordination modes for each MLH, ML and $MLH_{-1}$ species were found to have similar ground state energies, and therefore, all coordination modes have an equal probability of forming in solution. DFT calculated the $\lambda_{max}$ values of the MLH, ML and $MLH_{-1}$ species, but experimentally the absorption bands were not seen, which could be due to a combination of their low

concentration prevalence, or that they are hidden by the broad absorption bands of either the $MLH_{-2}$ species ($\lambda_{max}$ = 517–523 nm) and/or $Cu(H_2O)_6$ ($\lambda_{max}$ = 800 nm) [62].

The possible coordination modes for the MLH species (Figure 12) were the two coordination modes suggested from $^1$H NMR: coordination mode (a) has an amine-N and neighboring carbonyl-O coordination with the imidazole-N protonated; and coordination mode (b) has an imidazole-N and carboxyl-O coordination with the amine protonated. All coordination modes, except for coordination mode (b) of GLH and the coordination mode (b) of Sar-LH, became square pyramidal. The coordination mode (b) of GLH and Sar-LH became tetragonally distorted octahedral.

The possible coordination modes for the ML species of Cu-Sar-LH (Figure 13) were the two coordination modes proposed from potentiometry: coordination mode (a) has an amine-N, neighboring amide-N and carbonyl-O of leucine coordination with the imidazole-N protonated; and coordination mode (b) has an amine-N and neighboring amide-N coordination with the imidazole-N protonated. This agrees with the five lines of superhyperfine splitting that were found in EPR and indicated a $CuN_2O_2$ chromophore. Coordination modes (a) and (b) produced square planar and tetragonally distorted octahedral geometries, respectively.

The possible coordination modes for the $MLH_1$ species of Cu-GLH (Figure 14) were the three coordination modes proposed from potentiometry: coordination mode (a) has an amine-N and two amide-Ns coordination with a protonated imidazole-N; coordination mode (b) has an amine-N, neighboring amide-N and imidazole-N coordination; and coordination mode (c) has an amine-N, two amide-Ns and the carboxyl-O coordination with a protonated imidazole-N. Coordination modes (a) and (c) resulted in a square planar geometry, whereas coordination mode (b) resulted in a distorted square pyramidal geometry with an elongated axial water bond.

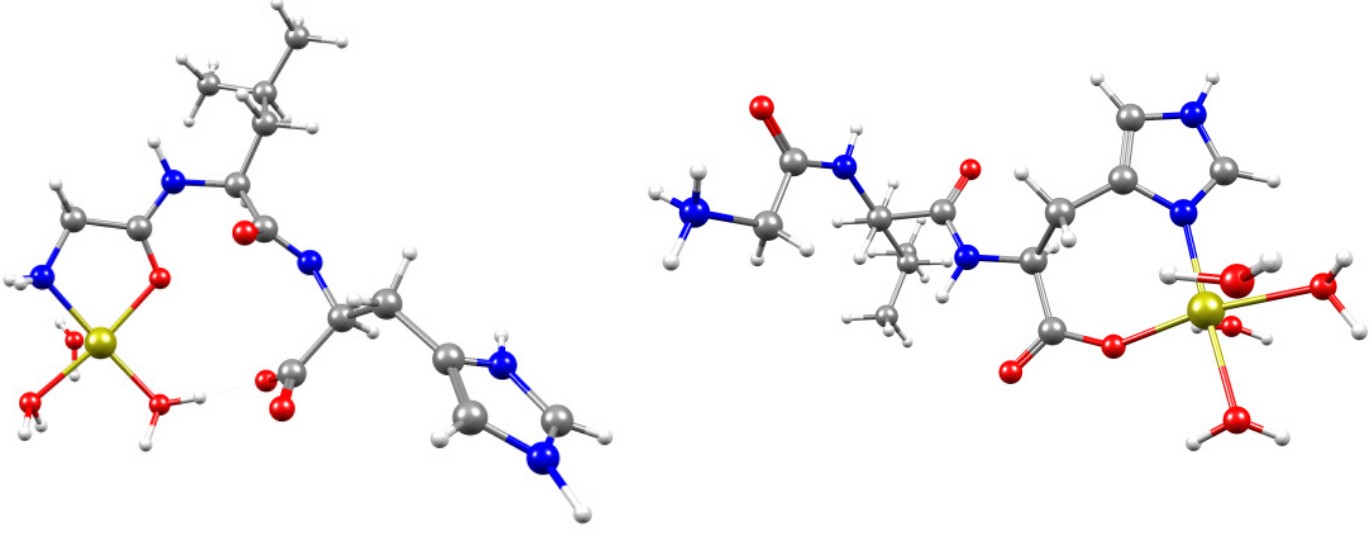

| Ligand: GLH coordination mode (**a**) | Ligand: GLH coordination mode (**b**) |
| Geometry: square pyramidal | Geometry: tetragonally distorted octahedral |
| $\lambda_{max}$: 710 nm | $\lambda_{max}$: 882 nm |

**Figure 12.** *Cont.*

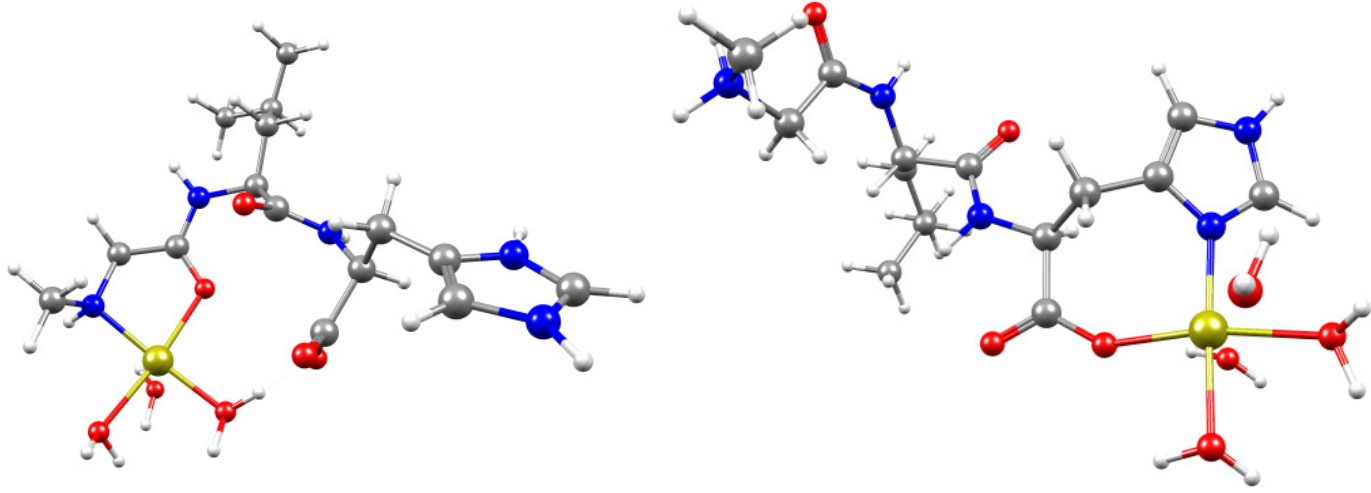

Ligand: Sar-LH coordination mode (**a**)

Geometry: square pyramidal

$\lambda_{max}$: 699 nm

Ligand: Sar-LH coordination mode (**b**)

Geometry: tetragonally distorted octahedral

$\lambda_{max}$: 889 nm

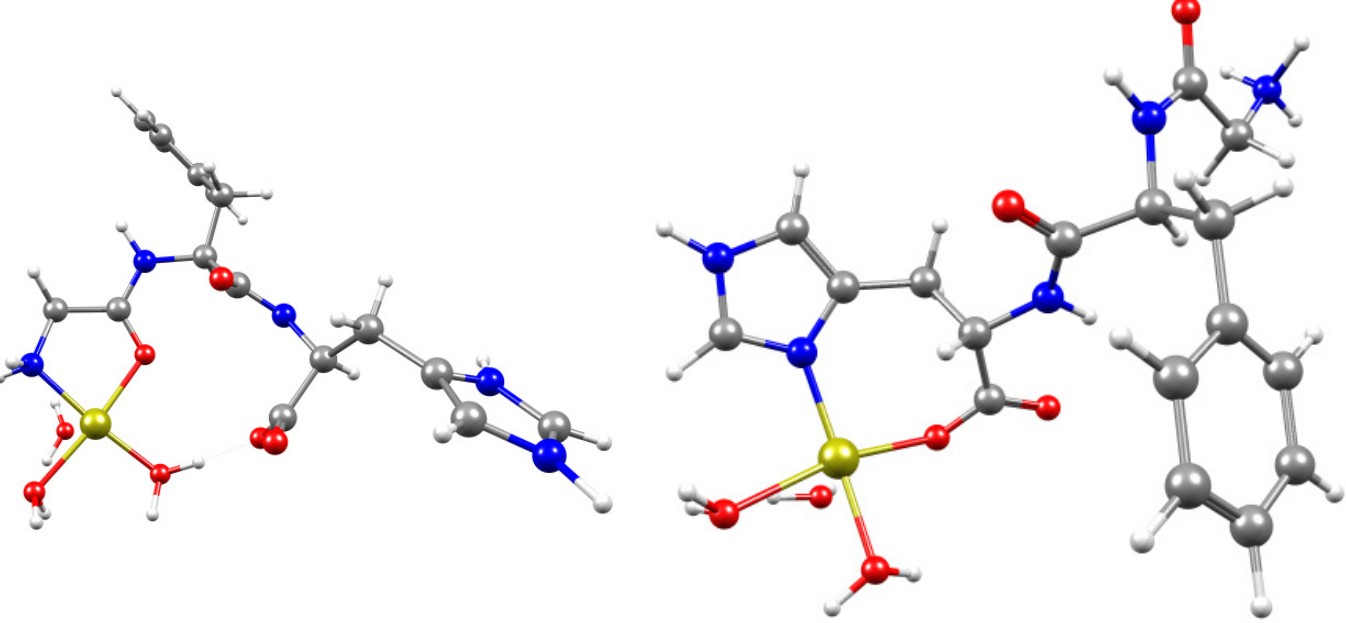

Ligand: GFH coordination mode (**a**)

Geometry: square pyramidal

$\lambda_{max}$: 837 nm

Ligand: GFH coordination mode (**b**)

Geometry: square pyramidal

$\lambda_{max}$: 875 nm

**Figure 12.** *Cont.*

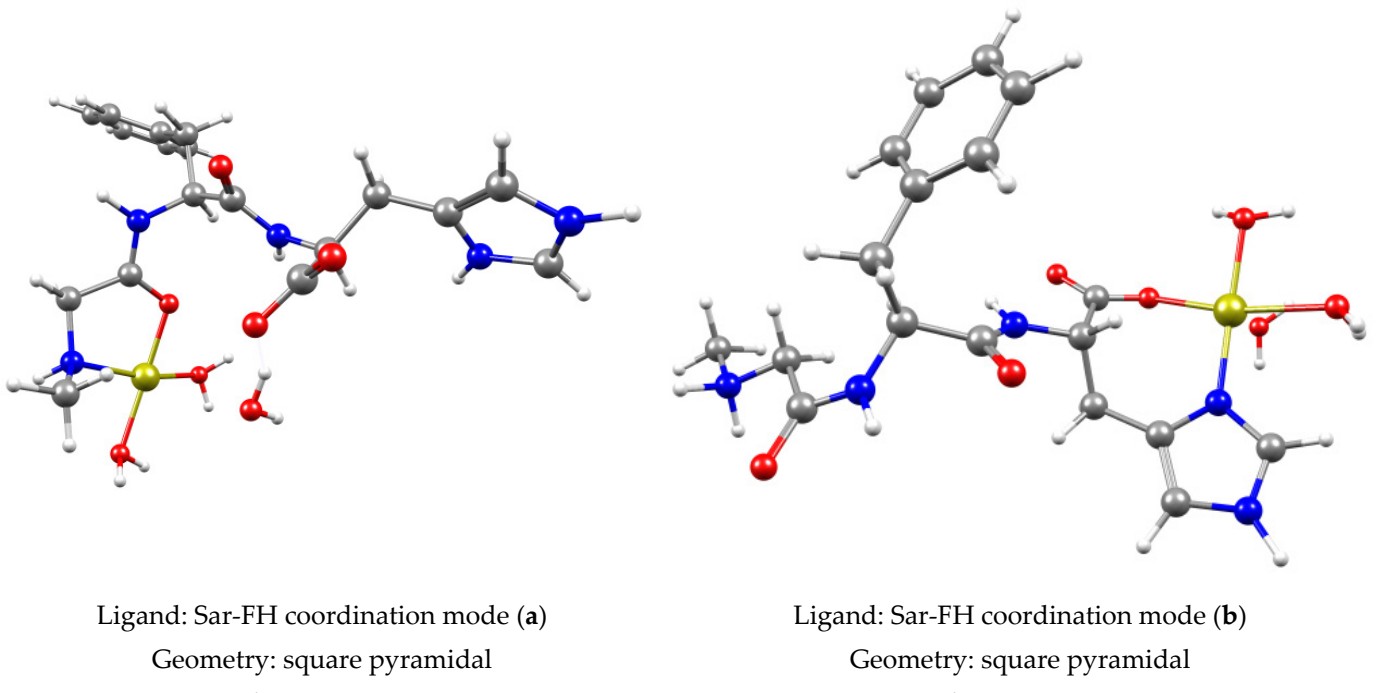

Ligand: Sar-FH coordination mode (**a**)

Geometry: square pyramidal

$\lambda_{\max}$: 707 nm

(**a**)

Ligand: Sar-FH coordination mode (**b**)

Geometry: square pyramidal

$\lambda_{\max}$: 868 nm

(**b**)

**Figure 12.** Visual representation of the proposed structures for the MLH species of (**a**) Cu-GLH, (**b**) Cu-Sar-LH, Cu-GFH and Cu-Sar-FH, as well as their geometries and calculated $\lambda_{\max}$ obtained at B3LYP/6-31++G** in water.

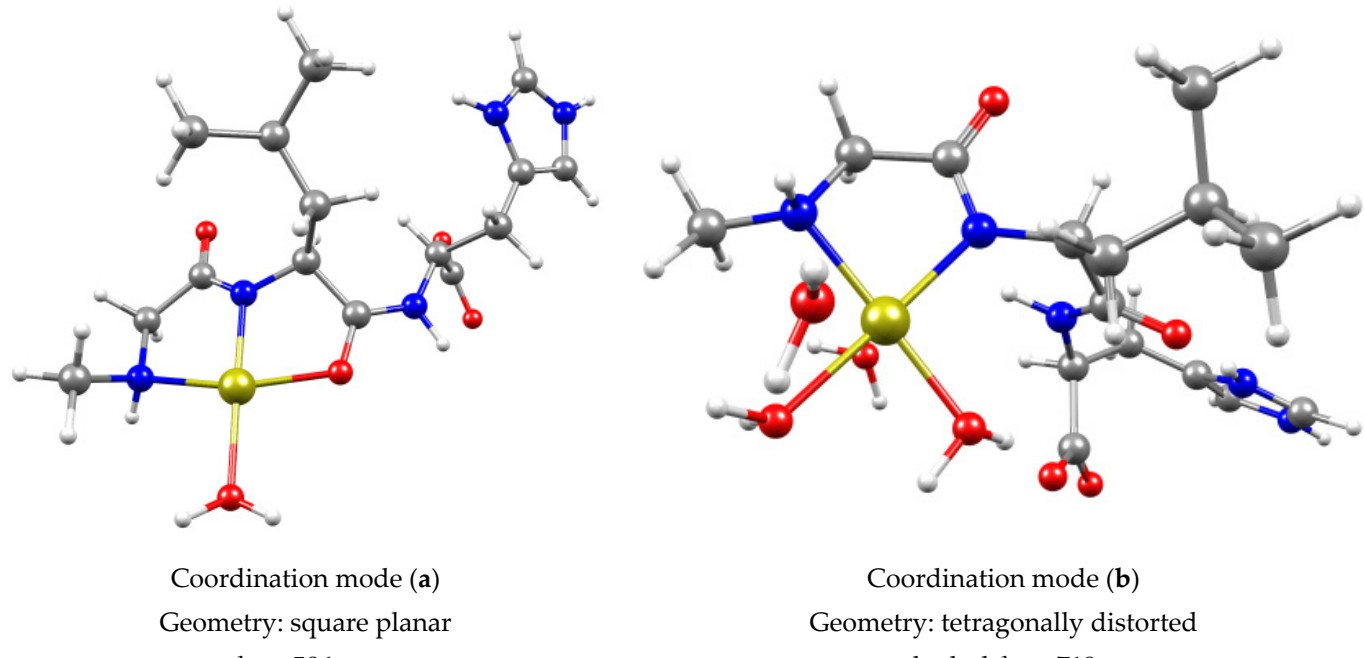

Coordination mode (**a**)

Geometry: square planar

$\lambda_{\max}$: 586 nm

Coordination mode (**b**)

Geometry: tetragonally distorted

octahedral $\lambda_{\max}$: 719 nm

**Figure 13.** Visual representation of the proposed structures for the ML species of (**a**) Cu-GLH, (**b**) Cu-Sar-LH, as well as their geometries and calculated $\lambda_{\max}$ obtained at B3LYP/6-31++G** in water.

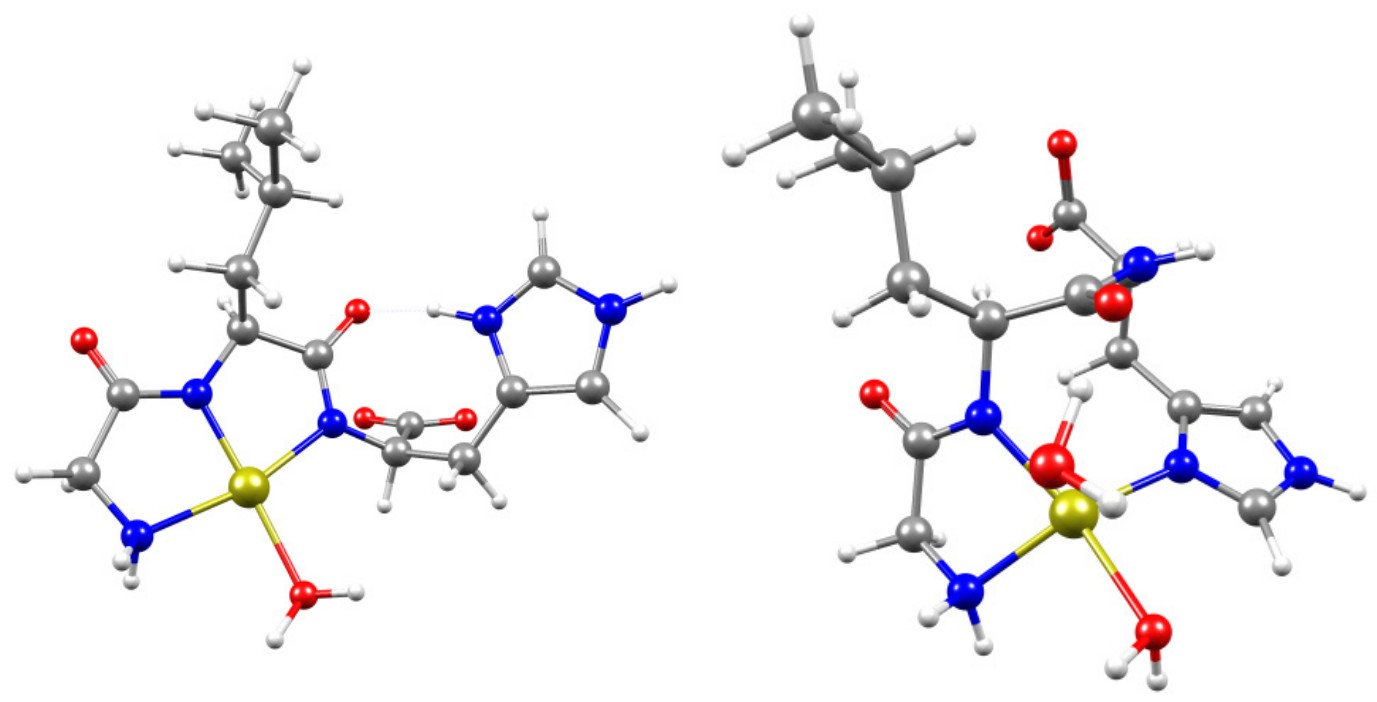

Coordination mode
Geometry: square planar
$\lambda$max: 569 nm

Coordination mode
Geometry: distorted square pyramidal
$\lambda$max: 763 nm

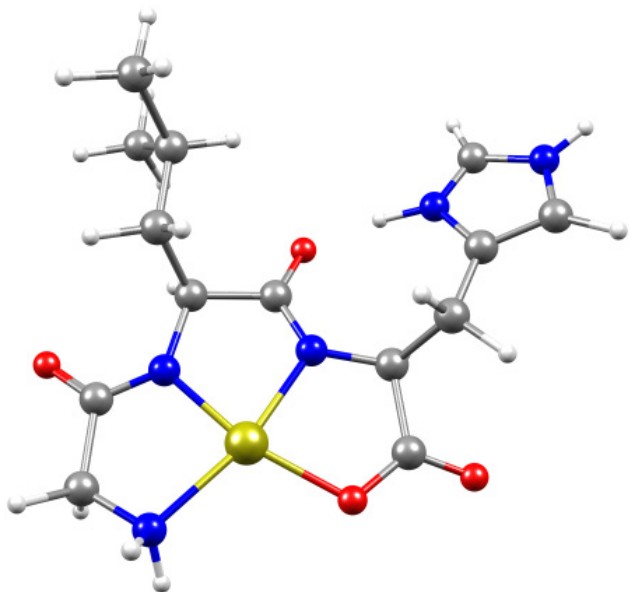

Coordination mode
Geometry: square planar
$\lambda$max: 592 nm

**Figure 14.** Visual representation of the proposed structures for the $MLH_{-1}$ species of Cu-GLH, as well as their geometries and calculated $\lambda_{max}$ obtained at B3LYP/6-31++G** in water.

The MLH$_{-2}$ species (Figure 15) is coordinated through the amine-N, two amide-Ns and the imidazole-N and the resultant structures all formed a square planar geometry. This is the exact structure and symmetry that was proposed by EPR and UV-vis. The high extinction coefficient, the low $g_{||}$ and high $A_{||}$, as well as the high value of the superhyperfine nitrogen constant, are all in agreement with the symmetry of this molecular modelling. Out of the four species, MLH$_{-2}$ is the only species where the experimental $\lambda_{max}$ values are known, and since the difference between the calculated and experimental $\lambda_{max}$ is 2–18 nm, it signifies that the selected DFT levels (B3LYP/6-31++G**) and solvent effect (SMD) are valid for this system [63,64].

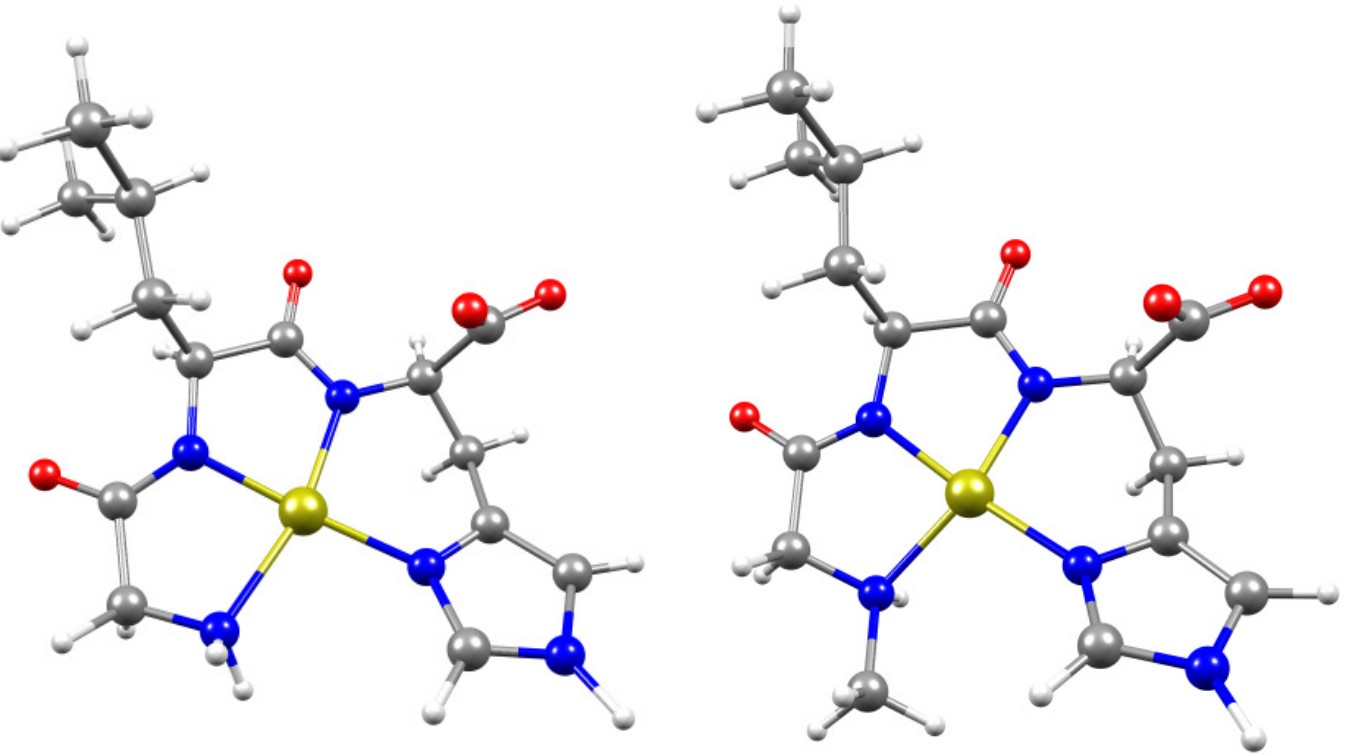

Ligand: GLH
Geometry: square planar
Calculated $\lambda_{max}$: 516 nm
Experimental $\lambda_{max}$: 518 nm

Ligand: Sar-LH
Geometry: square planar
Calculated $\lambda_{max}$: 541 nm
Experimental $\lambda_{max}$: 523 nm

**Figure 15.** *Cont.*

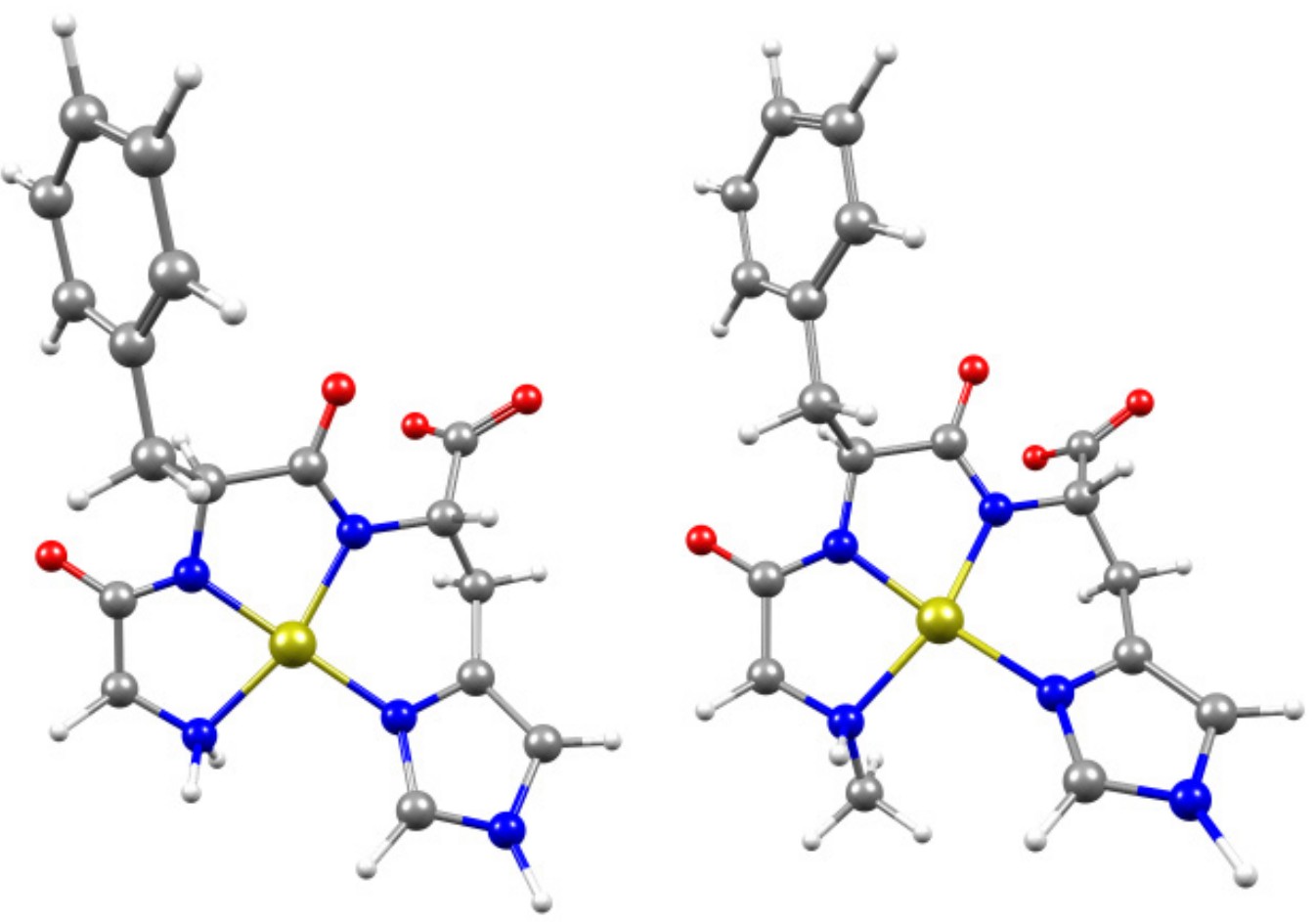

Ligand: GFH

Geometry: square planar

Calculated $\lambda_{max}$: 527 nm

Experimental $\lambda_{max}$: 517 nm

Ligand: Sar-FH

Geometry: square planar

Calculated $\lambda_{max}$: 534 nm

Experimental $\lambda_{max}$: 521 nm

**Figure 15.** Visual representation of the proposed structures for the MLH$_{-2}$ species of Cu-GLH, Cu-Sar-LH, Cu-GFH and Cu-Sar-FH, as well as their geometries, calculated $\lambda_{max}$ at B3LYP/6-31++G** in water and experimental $\lambda_{max}$.

## 3. Materials and Methods

### 3.1. Materials

All analytical grade chemicals and reagents were commercially available and were used as received. The ligands, glycyl-L-leucyl-L-histidine (GLH), sarcosyl-L-leucyl-L-histidine (Sar-LH), glycyl-L-phenylalanyl-L-histidine (GFH) and sarcosyl-L-phenylalanyl-L-histidine (Sar-FH), were purchased from GL Biochem (Shanghai, China). Purity was checked potentiometrically and by means of HPLC-MS and found to be >98%. Boiled Milli-Q water (18.2 MΩ.cm) was used to prepare all the potentiometric titration solutions in order to remove carbon dioxide, as outlined by Vogel [65]. The background electrolyte of the titration solutions was prepared with NaCl to have an ionic strength of 0.15 M so that it matched the ionic strength found in human blood [66].

Ligand solutions (5 mM) were prepared by dissolving weighed samples of GLH, Sar-LH, GFH and Sar-FH into a standardized hydrochloric acid and background electrolyte solution. Copper(II) solutions (0.01 M) were prepared using CuCl$_2 \cdot$2H$_2$O, adding the

background electrolyte to make the ionic strength 0.15 mol.dm$^{-3}$ and standardizing the solutions with EDTA.

### 3.2. Potentiometric Measurements

The potentiometric titrations were performed under an inert atmosphere of purified nitrogen gas at 25 °C and at a constant ionic strength of 0.15 mol.dm$^{-3}$ (NaCl) using a Metrohm 888 Titrando. The measurements took place in a double-walled titration vessel in a pH range of 2–11 and were kept at 25 ± 0.1 °C by a Haake thermostat bath (Thermo Fisher Scientific, Waltham, MA, USA). The amount of titrant added to the titrated solution was administered by a Metrohm 765 Dosimat automated burette (Metrohm, Herisau, Switzerland) via a capillary tip, which had a non-return valve. The amount of added titrant was controlled by the software program built into the Titrando, automatic titrator (Metrohm, Herisau, Switzerland), which also monitored the electromotive force. The titrated solution was stirred by a magnetic bar throughout the titration analysis. As an initial preparative setup, a range of Metrohm ion analysis pH buffers (pH 4, 7, and 9) was used to calibrate the slope of the electrode. Strong acid-strong base titrations (HCl/NaOH) were used to calculate the electrode potential, $E°$, and the dissociation constant of water, p$K_w$ [67,68]. The metal ligand ratios were 1:1, and both the protonation and metal ligand titrations were titrated against NaOH over a pH range of 2–11. The ESTA suite of programs [69] was then used to analyze the data from the potentiometric titrations. Potentiometric solutions were also analyzed spectroscopically on a Shimadzu UV-1800 recording spectrophotometer (Shimadzu, Kyoto, Japan) in the range from 200–800 nm.

### 3.3. Continuous Wave EPR Measurements

A Bruker Elexsys E500 CW-EPR spectrometer (Bruker, Billerica, MA, USA) driven by a PC running XEpr program under Linux and equipped with a Super-X microwave bridge operating at 9.3–9.9 GHz and an SHQE cavity was used throughout this work. All the frozen solution EPR spectra of copper(II) complexes were recorded in quartz tubes at 150 K by means of an ER4131VT variable temperature apparatus. The measurements at room temperature (RT) were recorded by means of a WG-812-H flat quartz cell, and occasionally a glass capillary was inserted into a quartz tube. In the case of RT EPR spectra, the isotropic magnetic parameters were evaluated from the average distances among the peaks of the experimental spectra recorded in the 2nd derivative mode.

EPR anisotropic magnetic parameters were obtained directly from the experimental EPR spectra, calculating them from the 2nd and the 3rd line to remove second order effects [70]. Perpendicular parameters were obtained by exploiting the appearance of the extra peak due to the angular anomaly, whose field can be used in connection with the parallel parameters to calculate with a certain accuracy $g_\perp$ and $A_\perp$, as explained in the literature [71,72].

Instrumental settings of the frozen solution EPR spectra were recorded as follows: number of scans 1–5 (in the case of RT spectra, more than 10 scans were occasionally required to collect an acceptable signal to noise ratio); microwave frequency 9.46–9.48 GHz; modulation frequency 100 kHz; modulation amplitude 0.2–0.6 mT; time constant 164–327 ms; sweep time 3–6 min; microwave power 10–20 mW; and linear receiver gain $1 \times 10^4$–$1 \times 10^5$. The instrumental settings of RT solution EPR spectra were substantially the same, except for the value of the microwave frequency. This was in the range of 9.70–9.80 GHz when using the flat quartz cell, and the microwave was powered up to 40 mW.

### 3.4. Preparation of Copper Complexes for EPR Measurements

Copper(II) complexes with these ligands were prepared by adding the appropriate amount of isotopically pure $^{63}$Cu(NO$_3$)$_2$ (50 mM) to an aqueous solution containing the pertinent ligand in slight excess. The absolute copper(II) concentrations ranged from 1–4 mM. Up to 10% methanol or glycerin was added to the aqueous solution containing the copper(II) complex species in order to increase the resolution of the low temperature

(LT) frozen solution spectra. The final aqueous solution pH was adjusted by means of an Orion 9103SC combined glass microelectrode, which was connected to an Orion Star A 211 pH meter. The pH was adjusted by using concentrated NaOH or $HNO_3$ as required.

### 3.5. Nuclear Magnetic Resonance (NMR)

For the $^1$H NMR spectra, 0.005 M solution for all ligands was prepared using 90% Milli-Q water and 10% $D_2O$. Tertiary butyl alcohol was added as an internal reference, and the pH was adjusted using a NaOH/HCl solution. The pH of each solution was recorded with an accuracy of 0.1 using a Crison micropH 2000 pH meter, which is equipped with a $\Omega$ metrohm glass electrode.

For the complexes, pH values were chosen in accordance with the species distribution diagrams. A copper(II) solution of 0.05 M was prepared using 90% Milli-Q water and 10% $D_2O$. The $^1$H NMR spectra were recorded on a Bruker 300 MHz spectrometer (Bruker, Billerica, MA, USA) and processed using Bruker Topspin software, version 4.0.7. The residual water peak was suppressed using excitation sculpting.

For the 1D Total Correlation Spectroscopy (TOCSY) NMR spectra , 0.001 g of GLH was weighed out and added to 0.9 mL of Milli-Q water and 0.1 mL of $D_2O$. A phosphate buffer was also added, and the pH was adjusted to 4.5 using NaOH/HCl. The TOCSEY spectra were recorded on a Bruker 600 MHz spectrometer (Bruker, Billerica, MA, USA) and processed using Bruker Topspin software, version 4.0.7.

### 3.6. Mass Spectrometry

Copper(II) complexes for each of the four ligands were prepared in 10 mL of Milli-Q water (18.2 M$\Omega$.cm), and the pH was adjusted to pH 5 with NaOH or HCl. The concentration of the ligands was 1 mM, and the concentration of copper(II) was adjusted to a slightly lower concentration of 0.7 mM to prevent precipitation. ESI-MS measurement samples were analyzed by direct infusion using the sample solvent as a carrier. Spectra were recorded on a Thermo TSQ quadrupole spectrometer (Thermo Fisher Scientific, Waltham, MA, USA) with a HESI ion source and analyzed with MS1 over a scan range of 100–600 $m/z$ in both the positive and the negative mode. The capillary temperature was 270 °C, and nitrogen was used as a nebulizing gas. The conditions for electrospray ionization were: spray voltage 3500 V, flow rate 5 µL/min, vaporizing temperature 100 °C, auxiliary gas pressure 10 (arbitrary units) and the sheath gas pressure 5 (arbitrary units). The data were viewed using the Thermos Qual Browser (Thermo Fisher Scientific, Waltham, MA, USA).

### 3.7. Density Functional Theory (DFT) Calculations

All calculations were performed using facilities provided by the University of Cape Town's High-Performance Computing center (hpc.uct.ac.za) using Gaussian 09 software (Gaussian, Pittsburgh, PA, USA) [73]. The coordination modes for the MLH, ML, MLH$_{-1}$ and MLH$_{-2}$ species of Cu-GLH, Cu-Sar-LH, Cu-GFH and Cu-Sar-FH were built using the demo Chemcraft program [74] with the appropriate multiplicity and charge of each structure. The starting geometry of copper(II) was also realigned to an octahedral geometry. The multiplicity for all structures is a doublet since copper(II) has an unpaired electron, and the charge for the MLH, ML, MLH$_{-1}$ and MLH$_{-2}$ species is +2, +1, 0 and −1, respectively. Each structure was then optimized at a B3LYP/6-31++G** level using water as a solvent. The solvent effect was implemented using the solvation model density (SMD) [75]. All optimized structures were found to be a minima with no imaginary frequency and were viewed in the demo Chemcraft program [74]. Time-dependent density functional theory (TD-DFT) calculations were conducted at the same level and solvent to obtain the electronic transitions of the copper(II) complexes. At first, 20 excited states were considered for excitation calculations; however, only the first 3 excited states were involved and considered for the rest of the calculations.

## 4. Conclusions

As an anti-inflammatory drug, the complexation of the copper(II) complexes cannot be too strong, or the ability to release copper(II) into the bloodstream is jeopardized. The stability constants belonging to the four Cu(II)-tripeptide systems have relatively low constants and, therefore, should be able to release copper(II) in vivo.

The objective to form a stable complex between copper(II) and the ligands, GLH, Sar-LH, GFH and Sar-FH, has been achieved. At the physiological pH, all four copper(II) complexes formed the $MLH_{-2}$ species, which is the major species in all systems. At low pH values, all four copper(II) complexes formed the MLH species, and Cu-GLH and Cu-Sar-LH also formed the $MLH_{-1}$ and ML species, respectively. The MLH, ML and $MLH_{-1}$ species have low concentrations and are thus minor species. The comparison between the thermodynamic stability of complexes with an N-methylated group and a non-N-methylated group was carried out. The expected increase in the stability of the complexes with the N-methylated group was not observed, and it was rationalized that the methyl group could either have steric effects or that there is a preference for the ammonium ions or charged amine groups to form hydrogen bonds with water.

In terms of structure determination, the absorption spectra in the visible region, as well as the EPR spectra recorded at room temperature and low temperature, suggested that the $MLH_{-2}$ species formed a stable $CuN_4$ chromophore with a square planar geometry at physiological pH values. The broadening of the $^1H$ NMR signals showed that the MLH species formed two simultaneous coordination modes in solution, where the one has coordinated to the amine-N and neighboring amide-N and the second has coordinated to the imidazole-N and carboxyl-O. The detection of the coordination mode for the $MLH_{-1}$ species was attempted by looking at the $pK_a$ value for the transition from $MLH_{-1}$ to $MLH_{-2}$ ($pK_a = 5.01$). This value was not close enough to the literature to confirm that a second amide is deprotonated nor coordinated to the imidazole-N and, thus, three probable coordination modes were proposed. The $pK_a$ value for the transition from the MLH to the ML species ($pK_a = 4.99$) is typical of a metal-assisted amide deprotonation, and thus two probable coordination modes were suggested, which consisted of a $CuN_2O_2$ chromophore. This chromophore was supported by the five lines of superhyperfine splitting in the EPR spectrum. The DFT calculations agreed with the experimental outputs that proposed the structure of $MLH_{-2}$ and showed that, for all proposed coordination modes for the MLH, ML and $MLH_{-1}$ species, each has an equal probability of forming simultaneously in solution. DFT calculations also provide an optimized structure for each coordination mode for each species and are viewed as the final structures.

**Supplementary Materials:** The following are available online at https://www.mdpi.com/article/10.3390/inorganics10010008/s1, Figure S1: $^1H$ NMR spectra of (a) GLH, (b) Sar-LH, (c) GFH and (d) Sar-FH at increasing pH values from 2–11. An arrow has been added to indicate the shifting of peaks over increasing pH values; Figure S2: The 1D selective gradient TOCSY NMR spectra (red) and $^1H$ NMR spectra (blue) of GLH at pH 4.5. (a) full spectrum of the irradiated amide-NH peak **d** at 8.246 ppm, (b) full spectrum of the irradiated amide-NH peak **i** at 8.511 ppm and (c) section of the spectrum of the irradiated amide-NH peak **i** at 8.511 ppm. An arrow has been added to indicate the irradiated amide-N; Figure S3: The $^1H$ NMR spectra (blue) of the ligand GFH and the $^1H$ NMR spectra (red) after GFH has been titrated with copper(II) to reach a 5:1 ligand copper(II) ratio at a pH of 4.8 in 90% water and 10% $D_2O$. $^1H$ NMR spectra (green) with arrows pointing to the significant broadening of peaks **b** and **b′**, after GFH has been titrated with copper(II); Figure S4: The 1H NMR spectra (blue) of the ligand Sar-FH and the 1H NMR spectra (red) after Sar-FH has been titrated with copper(II) to reach a 5:1 ligand copper(II) ratio at a pH of 4.8 in 90% water and 10% D2O. 1H NMR spectra (green) with arrows pointing to the significant broadening of peaks b and b′, after Sar-FH has been titrated with copper(II). Table S1: Structural assignments of m/z base peaks that were found in the ESI-MS spectrum for Cu-GLH, Cu-Sar-FH and Cu-GFH at pH 5 (positive mode) with a 1:1 ratio and concentration of 1 mM for GLH, Sar-FH and GFH, and 0.7 mM for copper(II) in aqueous solution.

**Author Contributions:** Conceptualization, G.M.V. and S.A.B.; formal analysis, G.E.J., G.M.V., A.N.H., R.A., R.P.B. and G.V.; investigation, G.M.V., A.N.H. and G.V.; methodology, G.E.J. and R.P.B.; project administration, G.E.J.; resources, G.E.J.; supervision, G.E.J., A.N.H. and S.A.B.; visualization, G.M.V. and R.A.; writing—original draft, G.M.V.; writing—review and editing, G.M.V., R.A. and S.A.B. All authors have read and agreed to the published version of the manuscript.

**Funding:** This research was funded by the National Research Foundation of South Africa (grant Nos 93450 and 85466 to G.E.J., and bursary to G.M.V.) and the University of Cape Town Research Committee. The Centre for High-Performance Computing (CHPC), South Africa, provided computational resources for this research project.

**Institutional Review Board Statement:** Not applicable.

**Informed Consent Statement:** Not applicable.

**Data Availability Statement:** Not applicable.

**Conflicts of Interest:** The authors declare no conflict of interest. The funders had no role in the design of the study; in the collection, analyses, or interpretation of data; in the writing of the manuscript; or in the decision to publish the results.

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
