# Peer review of "Aqueous Solution Equilibria and Spectral Features of Copper Complexes with Tripeptides Containing Glycine or Sarcosine and Leucine or Phenylalanine"

_inorganics, doi:10.3390/inorganics10010008_

Round 1
Reviewer 1 Report
The paper is focused on spectral study of the tripeptides with copper(II). The study in solution is well combined with DFT calculations. I would like to recommend this work for acceptance after minor revision. The comments to the specific part of the text are listed below.
Line 25: have potential anti-inflammatory (double space between potential and anti-inflammatory) => have potential anti-inflammatory
Line 39: Is there any reason for numbers in Keyword part?
Table 1: format of the table could be improved for complex pqr parameters (shifts of number due to negative magnitude)
Figure 1: It is necessary to have this fig together (not end and start of pages!); Maybe, it will be nice to have for each fig1a-d the name of the ligand in picture (faster orientation)
Figure 5: It is necessary to have this fig together (not end and start of pages!)
Figure 9: It is necessary to have this fig together (not end and start of pages!)
Figure 10: I would recommend to show only on NMR spectrum in main text and the rest can be found in SI.
Table 4: It can be reduced to confirm the presence of the complex (just one) with uncomplexed ligand. The rest of the data is better toad into SI.
Line 511: Is it Figure 1 or 14?
Figures 12-15: There is lot of the data, I would recommend to add some of these data to put into SI and the important structures show in main text.
Author Response
"Please see the attachment."

Reviewer 2 Report
The paper submitted by prof. Graham Jackson and collaborators concerns the study of a series of copper (II) complexes bearing tripeptide ligands based on different combination of glycine sarcosine leucine and phenylalanine. The physicochemical properties of four different compounds susceptible to exhibiting anti-inflammatory activity are presented.
This work involves detailed potentiometric measurements at different pH values. UV visible spectroscopy coupled with EPR spectroscopy measurements have also been performed to determine the structure of the different species. The different binding sites have been determined from an interesting 1H NMR study, all these analyses being completed by masss spectroscopy measurements. Spectroscopic analyses results are well supported by DFT calculations that validates experimental conclusions.
Overall, this study is very well presented. The experimental data are complete and rigorously displayed. However the quality of the article could be enhanced. Indeed, the introduction could be updated by inserting more recent examples of copper complexes involving polypeptide ligands, even if the latter do not concern the same objectives, namely species with anti-inflammatory properties. Nevertheless, this article will deserve to be published in inorganics journal after minor corrections.
Page 21 : Figure 14 instead of Figure 1.
Author Response
"Please see the attachment.
